# Teamwork makes von Neumann work: Min-Max Optimization in Two-Team Zero-Sum Games

## Abstract

Motivated by recent advances in both theoretical and applied aspects of multi-player games, spanning from e-sports to multi-agent generative adversarial networks, we focus on min-max optimization in team zero-sum games. In this class of games, players are split into two teams with payoffs equal within the same team and of opposite sign across the opponent team. Unlike the textbook two-player zero-sum games, finding a Nash equilibrium in our class can be shown to be CLS-hard, i.e., it is unlikely to have a polynomial-time algorithm for computing Nash equilibria. Moreover, in this generalized framework, we establish that even asymptotic last iterate or time average convergence to a Nash Equilibrium is not possible using Gradient Descent Ascent (GDA), its optimistic variant, and extra gradient. Specifically, we present a family of team games whose induced utility is non-multilinear with non-attractive *per-se* mixed Nash Equilibria, as strict saddle points of the underlying optimization landscape. Leveraging techniques from control theory, we complement these negative results by designing a modified GDA that converges locally to Nash equilibria. Finally, we discuss connections of our framework with AI architectures with team competition structures like multi-agent generative adversarial networks.

## 1 Introduction

*Team competition* has played a central role in the development of Game Theory (Marschak, 1955; von Stengel & Koller, 1997; Bacharach, 1999; Gold, 2005), Economics (Marschak, 1955; Gottinger, 1974) and Evolutionary Biology (Nagylaki, 1993; Nowak et al., 2004), however, the behavior of the underlying dynamics within the teams are usually sidelined. Either for reasons of mathematical convenience or bigger picture understanding, "teams" in literature are typically modeled as if they were unitary actors, i.e., single individuals without unveiling the internal decision-making of the team members (see Kim et al. (2019)).

For instance, in the biology setting of *weak selection model* (Nagylaki, 1993; Chastain et al., 2014; Mehta et al., 2015) species are modeled to compete as teams, while at the crux of the matter the genes of each species are the actual players and their alleles are the actions in the survival game. Similarly, it was the social-media collaboration of the Reddit retail trading crowd as a team that touches off the last year's GameStop frenzy of short squeeze (Umar et al., 2021; Hasso et al., 2021) transforming the markets into a tug of war game against the team of Wall Street hedge funds.

Recently, these intrinsic details behind the competition among teams have attracted renewed interest in the Machine Learning community, motivated by the advent of multi-agent systems that are used for generative tasks or playing complex games like CTF (Jaderberg et al., 2019) or Starcraft (Vinyals et al., 2019). So as to win this kind of games, self-training AI systems have to develop both collaborative attributes (coordination within each team) as well as contesting ones (competition across the teams). Moreover, following the complementary thread of multi-agent generative adversarial network research, the creation of a pool by efficient incumbent agents, either in generators (Arora et al., 2017; Hoang et al., 2017; 2018; Zhang et al., 2018; Tang, 2020), or discriminators (Hardy et al., 2019; Albuquerque et al., 2019) has been tested providing significant statistical and computa-

tional benefits. In this direction, researchers strive to harness the efficacy of distributed processing, utilizing shallower networks that can learn all the while more diverse datasets [1].

In order to shed some light on this persistent strain of research, the main premise of the theoretical scaffolding developed in this paper is that

> *The "unitary two-players" min-max approach misses the critical component*
> *of the collective strategy making within each competing team.*

**Our class of games.** In this regard, we turn our attention to *Two-Team Zero-Sum games*, proposed by Schulman & Vazirani (2019b), a quite general class of min-max optimization problems that include bilinear games as well as a wide range of non-convex non-concave games. In this class, the players fall in two teams of size $k_1, k_2$ and submit their own probabilistic strategy vector independently, akin to a general normal form multi-player game. Following the econometric common value assumption of Marschak (1955), what makes a group of players a *team* is that in any outcome the players of each team receive an identical payoff. Thus, to build some intuition, it is easy to see that if perfect coordination existed within each team, the interaction between the teams is merely a zero-sum game between two "virtual" players. To streamline our presentation here, we defer the more precise description of our model to Section 2.

**Challenges behind Two-Team Zero-Sum games.** In the archetypical case of two players, i.e., $(k_1 = k_2 = 1)$, min-max strategies are typically thought of as the axiomatically correct predictions thanks to the seminal Von Neumann's minmax theorem (Von Neumann, 1928). Unfortunately, min-max optimization for case $k > 1$ is a much more tenuous affair: Schulman & Vazirani (2019b) preclude the existence of unique value by presenting a family of team games where $\min \max \neq \max \min$ together with bounds about this duality gap, which quantifies exactly the effect of exchanging the order of strategy commitment either between the teams or the players thereof.

If defining the correct figure of merit for Team games is rife with frustration, what is even more demanding is understanding what kind of algorithms/dynamics are able to solve this problem when a game-theoretically meaningful solution exists: Firstly, computing local Nash Equilibria (NE) in general non-convex non-concave games is PPAD-complete (Daskalakis et al., 2009; 2021). Thus, all well-celebrated first-order methods, like gradient descent-ascent (Lin et al., 2020; Daskalakis & Panageas, 2019), its optimistic (Popov, 1980; Daskalakis & Panageas, 2018; Mertikopoulos et al., 2019) and extra gradient variant (Korpelevich, 1976) would require an exponential number of steps in the parameters of the problem to find an appoximate NE under Nemirovsky-Yudin (Nemirovskij & Yudin, 1983) oracle optimization model. Secondly, even if a regret notion could be defined, no-regret methodology is guarranteed to attract only to the set of coarse correlated equilibria (CCE) (Fudenberg, 1991; Hannan, 2016; Flokas et al., 2020; Giannou et al., 2021), a weaker notion that may be exclusively supported on strictly dominated strategies, even for simple symmetric two-player games (See also Viossat & Zapechelnyuk (2013)).

Whilst the aforementioned intractability failures for the general case of non-convex non-concave min-max problems provides significant insights, they can not *a fortiori* answer the fundamental question, restricted in the model of Two-Team Zero-Sum Games:

> *Can we compute Nash equilibria in Two-Team Zero-Sum Games and*
> *ultimately are there first-order methods that converge to them under tangible guarantees?*

**Our results.** To the best of our knowledge, the following contributions are the first-of-its-kind type of results for the case of Two-Team Zero-Sum games:

- For the case of the computational complexity of approximate (possibly mixed) NE we establish a sweeping negative result proving that it is CLS-hard (Theorem 3.1), i.e., is computationally harder than finding pure NE in a congestion game or finding approximate gradient descent fixed points.
- From an optimization perspective, we settle these questions with a resounding "no" for all the well-known discrete gradient flow variations. Specifically, we present a simple family of two-team with two-players zero-sum games where Projected-GDA, Optimistic-GDA, and Extra Gradient fail even to stabilize around a mixed NE, when they are initialized nearby (Theorem 3.5).

---

[1] Indeed, from the training perspective, it's more computationally preferable to back-propagate through two equally sized neural networks with smaller capacity rather than through a giant single one that would be twice as deep.(Tang, 2020)

Additionally, for the category GDA in the non-degenerate team games with unique mixed NE, one could acquire an even stronger result for any high-dimensional configuration of actions and players. (Theorem 3.2)

- In order to make some substantial headway under the burden of the above instability results, we shift our attention to adaptive control generalizations of the celebrated Washout filters–traditionally used for stabilizing the Dutch-roll motion of an aircraft during a flight (Hassouneh et al., 2004; Grant & Reid, 1997). Inspired by this framework, we propose the modified KPV-GDA[2] which consists of a tandem combination of GDA together with a stabilizing feedback introduces by Bazanella et al. (1997).

$$\begin{cases} state^{(k+1)} = & state^{(k)} + \eta GDA(state^{(k+1)}) + \eta K(state^{(k)} - stress^{(k)}) \\ stress^{(k+1)} = & stress^{(k)} + \eta P(state^{(k)} - stress^{(k)}) \end{cases} \quad \text{(KPV-GDA)}$$

The main linchpin of KPV-GDA method is the Simon's and Theil's (Simon, 1956; Theil, 1957) *certainty equivalence principle*, a widely used methodology in Control theory in developing applied dynamic rational expectations models. According to this principle, the feedback law is split into two optimization steps whereby $K-step$ attracts quickly the $state$ to the $stress$ while $P-step$ converges slowly to the fixed points of GDA. Compared with the plethora of the proposed dynamics for min-max problems, the crucial advantage of the afore-described technique is that does not introduce any extra fixed points than GDA's ones. In Section 2.2, we provide some illustrative examples of KPV-GDA technique, while in Theorem 3.7 we prove the existence of such control feedback for our class of games.

- Finally, in Section 4 we provide a series of experiments in simple two-team zero-sum games showcasing both the messy behaviors of traditional methods like GDA,OGDA and the power of KPV-GDA method in these optimization environments. Additionally, we show that multi-agent GAN architectures achieve better performance than the single-agent ones, in terms of network capacity, when they are trained in synthetic or real-world datasets like CIFAR10.

## 2 PRELIMINARIES

### 2.1 DEFINITIONS

**Our setting.** Formally, a *two-team game* in normal form is defined as a tuple $\Gamma = \Gamma(\mathcal{N}, \mathcal{A}, u)$ consisting of $(i)$ a finite set of *players* $\mathcal{N}$, split into two teams $A, B$ with $k_A$ and $k_B$ players correspondingly such that: $\mathcal{N} = \mathcal{N}_A \cup \mathcal{N}_B = \{A_1, \cdots, A_{k_A}, B_1, \cdots, B_{k_B}\}$; $(ii)$ a finite set of *actions* (or *pure strategies*) $\mathcal{A}_i = \{\alpha_1, \ldots, \alpha_{n_i}\}$ per player $i \in \mathcal{N}$; $(iii)$ each team's *payoff* function $u_A, u_B : \mathcal{A} \to \mathbb{R}$, where $\mathcal{A} := \prod_i \mathcal{A}_i$ denotes the ensemble of all possible action profiles $\alpha = (\alpha_{A_1}, \ldots, \alpha_{A_{k_A}}, \alpha_{B_1}, \ldots, \alpha_{B_{k_B}})$ while the *individual utility* of a player is identical to her teammates, i.e., $u_i = u_A$ & $u_j = u_B \ \forall (i,j) \in \mathcal{N}_A \times \mathcal{N}_B$. In this general context, players could also adhere *mixed strategies*, i.e, probability distributions $s_k \in \Delta(\mathcal{A}_k)$ over the pure strategies $\alpha_k \in \mathcal{A}_k$. Correspondingly, we define the product distributions $\mathbf{x} = s_{A_1} \otimes \cdots \otimes s_{A_{k_A}}, \mathbf{y} = s_{B_1} \otimes \cdots \otimes s_{B_{k_B}}$ as the teams' strategies. Collectively, we will write $\mathcal{X} := \prod_{i \in \mathcal{N}_A} \mathcal{X}_i = \prod_{i \in \mathcal{N}_A} \Delta(\mathcal{A}_i), \mathcal{Y} := \prod_{i \in \mathcal{N}_A} \mathcal{Y}_i = \prod_{i \in \mathcal{N}_B} \Delta(\mathcal{A}_i)$ the space of mixed strategy profiles of teams $A, B$.

Similarly with the bilinear two-player games, the teams' utility functions can be expressed via the payoff-*tensors* $\mathbf{A}, \mathbf{B} \in \mathbb{R}^\tau$ with $\tau = \prod_{i \in \mathcal{N}} |\mathcal{A}_i|$ and acquire the form[3]:

$$u_A = \mathbf{A}_{\mathbf{x}}^{\mathbf{y}} \ \& \ u_B = \mathbf{B}_{\mathbf{x}}^{\mathbf{y}} \quad (2.1)$$

---

[2]The name "KPV-GDA" is an initialism from $(K, P)$-Vaned Gradient Descent Ascent method; Just like the tail section of an aircraft where the vanes are flight control surfaces that control the unstable yaw, $(K, P)$ control feedback aims to stabilize the unstable arrows of Gradient flow around a mixed NE.

[3]Figuratively, the latter form denotes what is known as a tensor contraction given: $\mathbf{A}_{\mathbf{x}}^{\mathbf{y}} = \sum_{i,\ldots,j,k\ldots,l} x_{i,\ldots,j} A_{i,\ldots,j,k\ldots,l} y_{k\ldots,l}$ If $\mathbf{x}, \mathbf{y}$ have shapes $(i,j)$ and $(k,l)$ that would be equivalent to `u = einsum('ijkl,ij,kl', A, x, y)`

In terms of solutions, we focus on the per player *Nash Equilibrium* (NE), i.e., a state strategy profile $s^* = (\mathbf{x}, \mathbf{y}) = \left( (s^*_{A_1}, \ldots, s^*_{A_{k_A}}), (s^*_{B_1}, \ldots, s^*_{B_{k_B}}) \right)$ such that

$$u_i(s^*) \geq u_i(s_i; s^*_{-i})^4 \text{ for all } s_i \in \Delta(\mathcal{A}_i) \text{ and all } i \in \mathcal{N} \tag{NE}$$

The state strategy profile $s^*$ is called *pure* if every player of both teams chooses a single action; otherwise we say that it is mixed. Finally, a two-team game is called *two-team zero-sum* if $u_A = -u_B$ or equivalently $\mathbf{A} + \mathbf{B} = \mathbf{O}$.

*Remark* 2.1. A quite technical prerequisite for the rest of this work, we will assume that a succinct representation of the utility tensors of the game is available or equivalently that a payoff oracle provides efficiently both the value of the utility function and its derivatives for a specific input, which is consistent with the vast majority of the applications that are described in the literature (von Stengel & Koller, 1997).

**A first approach on computing Nash equilibria in Two-Team Zero-Sum games.** Given the existence of the duality-gap between the $\min\max$ and $\max\min$, in lieu of the two-player zero-sum game, an equilibrium in our setting can not be computed via linear programming. For the goal of computing Nash equilibria in two-team zero-sum games, we have experimented with a selection of first-order methods that have been utilized with varying success in the setting of the two-person zero-sum case. Namely, we analyze the following methods: *i) Gradient Descent-Ascent ii) Optimistic Gradient Descent-Ascent iii) Extra Gradient Method iv) Optimistic Multiplicative Weights Update Method* We defer their precise definitions in Appendix B. The below remark will play a key role in the sequel.

*Remark* 2.2. Any fixed point of the aforementioned discrete-time dynamics on the utility function corresponds necessarily to the Nash equilibria of the game.

Hence, an important testbed for the long-run behavior of GDA, OGDA, and EG methods is to examine whether these methods stabilize around their fixed points, which effectively constitute the Nash equilibria of the game. In Section 3.2, we show that in lack of pure Nash equilibria, all the above methods fail to stabilize on their fixed points even for a simple class of $(2, 2)$-players game, and as a consequence to the mixed Nash equilibria of the game.

The presence of these results showcases the need for a different approach that lies outside purely optimization-based ideas. Inspired by the applications of washout filters to stabilize highly susceptible systems and their adaptive control generalizations, we design a new incarnation of GDA vaned by two matrices-control feedback. Surprisingly, in contrast with the aforementioned traditional methods, our proposed technique accomplishes last-iterate stabilization on its fixed point, i.e., the mixed Nash equilibria of the team game.

$(K, P)$**-Vaned GDA Method.** After concatenating the vectors of the minimizing and the maximizing agents $\mathbf{z}^{(k)} = (\mathbf{x}^{(k)}, \mathbf{y}^{(k)})$ we can write our method, for appropriate matrices $K, P$:

$$\begin{cases} \mathbf{z}^{(k+1)} = & \Pi_{\mathcal{Z}} \left\{ \mathbf{z}^{(k)} + \eta \left( \begin{smallmatrix} -\nabla_{\mathbf{x}} f(\mathbf{z}^{(k)}) \\ \nabla_{\mathbf{y}} f(\mathbf{z}^{(k)}) \end{smallmatrix} \right) + \eta K (\mathbf{z}^{(k)} - \boldsymbol{\theta}^{(k)}) \right\} \\ \boldsymbol{\theta}^{(k+1)} = & \Pi_{\mathcal{Z}} \left\{ \boldsymbol{\theta}^{(k)} + \eta P (\mathbf{z}^{(k)} - \boldsymbol{\theta}^{(k)}) \right\} \end{cases} \tag{2.2}$$

Intuitively, the added variable $\boldsymbol{\theta}^{(k)}$ holds an estimate of the fixed point, and through the feedback $\eta K (\mathbf{z}^{(k)} - \boldsymbol{\theta}^{(k)})$ the vector $\mathbf{z}$ stabilizes around that estimate which slowly moves towards the real fixed point of the plain GDA dynamic. It is crucial to note that no additional fixed points are introduced to the system.

## 2.2 Two Illustrative Examples

Our first example supports a double role: Firstly, it exemplifies how our two-team min-max competition can capture the formulation of multi-agent GANs' architectures. Secondly, it hints also at an early separation between the optimization methods, since as we will see GDA will not converge to the Nash Equilibrium/ground-truth distribution.

---

[4]We are using here the standard shorthand $(s_1, \ldots, s_i, \ldots, s_{|\mathcal{N}|})$ to highlight the strategy of a given player $i \in \mathcal{N}$ versus the rest of players $\mathcal{N} \setminus \{i\}$.

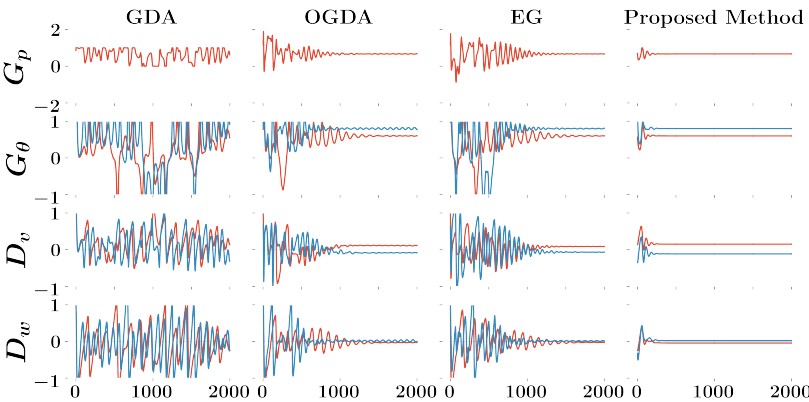

Figure 1: Parameter training of the configuration under different algorithms

### 2.2.1 LEARNING A MIXTURE OF GAUSSIANS WITH MULTI-AGENT GANS

Consider the case of $\mathcal{M}$, a mixture of gaussian distribution with two components, $C_1 \sim \mathcal{N}(\boldsymbol{\mu}, \boldsymbol{I})$ and $C_2 \sim \mathcal{N}(-\boldsymbol{\mu}, \boldsymbol{I})$ and mixture weights $\pi_1, \pi_2$ to be positive such that $\pi_1 + \pi_2 = 1$ and $\pi_1, \pi_2 \neq \frac{1}{2}$.

To learn the distribution above, we utilize an instance of a *Team*-WGAN in which there exists a generating team of agents $G_p : \mathbb{R} \to \mathbb{R}, G_{\boldsymbol{\theta}} : \mathbb{R}^n \to \mathbb{R}^n$, and a discriminating team of agents $D_{\mathbf{v}} : \mathbb{R}^n \to \mathbb{R}, D_{\mathbf{w}} : \mathbb{R}^n \to \mathbb{R}$, all described by the following equations:

$$\text{Generators:} \quad G_p(\zeta) = p + \zeta \ , \ G_\theta(\mathbf{z}) = \ \mathbf{z} + \boldsymbol{\theta}$$
$$\text{Discriminators:} \ D_{\mathbf{v}}(\mathbf{y}) = \langle \mathbf{v}, \mathbf{y} \rangle \ , D_{\mathbf{w}}(\mathbf{y}) = \sum_i w_i y_i^2 \tag{2.3}$$

The generating agent $G_\theta$ maps random noise $\mathbf{z} \sim \mathcal{N}(0, \boldsymbol{I})$ to samples while generating agent $G_p(\zeta)$, utilizing an independent source of randomness $\zeta \sim \mathcal{N}(0, 1)$, probabilistically controls the sign of the output of the generator $G_\theta$. The probability of ultimately generating a sample $\mathbf{y} = \mathbf{z} + \boldsymbol{\theta}$ is equal to $\zeta + p$, while the probability of the sample being $\mathbf{y} = -\mathbf{z} - \boldsymbol{\theta}$ is equal to $1 - (p + \zeta)$.

On the other end, there stands the discriminating team of $D_{\mathbf{v}}, D_{\mathbf{w}}$. Discriminators, $D_v(\mathbf{y}), D_w(\mathbf{y})$ map any given sample $\mathbf{y}$ to a scalar value accounting for the realness or fakeness of it – negative meaning fake, positive meaning real. The discriminators are disparate in the way they measure the realness of samples as seen in their definitions.

We follow the formalism of the Wasserstein GAN to form the optimization objective:

$$\max_{\mathbf{v}, \mathbf{w}} \min_{\boldsymbol{\theta}, p} \left\{ \begin{array}{c} \mathbb{E}_{\mathbf{y} \sim real} \Big[ D_{\mathbf{v}}(\mathbf{y}) + D_{\mathbf{w}}(\mathbf{y}) \Big] \\ - \\ \mathbb{E}_{z \sim \mathcal{N}(0, \boldsymbol{I}), \zeta \sim \mathcal{N}(0,1)} \left[ \left[ \begin{array}{c} G_p(\zeta) \cdot \Big( D_{\mathbf{v}}\big(G_{\boldsymbol{\theta}}(\mathbf{y})\big) + D_{\mathbf{w}}\big(G_{\boldsymbol{\theta}}(\mathbf{y})\big) \Big) \\ + \\ \big(1 - G_p(\zeta)\big) \cdot \Big( D_{\mathbf{v}}\big(-G_{\boldsymbol{\theta}}(\mathbf{y})\big) + D_{\mathbf{w}}\big(-G_{\boldsymbol{\theta}}(\mathbf{y})\big) \Big) \end{array} \right] \right] \end{array} \right\} \tag{2.4}$$

Equation equation 2.4 yields the simpler form:

$$\max_{\mathbf{v}, \mathbf{w}} \min_{\boldsymbol{\theta}, p} (\pi_1 - \pi_2)\mathbf{v}^T \boldsymbol{\mu} - 2p\mathbf{v}^T \boldsymbol{\theta} + \mathbf{v}^T \boldsymbol{\theta} + \sum_i^n w_i(\mu_i^2 - \theta_i^2) \tag{2.5}$$

It is easy to check that Nash equilibria of Equation equation 2.4 must satisfy:

$$\left\{ \begin{array}{llll} \boldsymbol{\theta} & = & \boldsymbol{\mu}, & p = 1 - \pi_2 = \pi_1 \\ \boldsymbol{\theta} & = & -\boldsymbol{\mu}, & p = 1 - \pi_1 = \pi_2. \end{array} \right\}$$

Figure 1 demonstrates both GDA's failure and OGDA, EG, and our KPV-GDA method's success to converge to the above Nash equilibria and simultaneously to discover the groud truth mixture.

### 2.2.2 MULTIPLAYER MATCHING PENNIES

Interestingly enough, there are non-trivial instances of two-team competition settings that even Optimistic GDA and EG Method fail to converge. Such is the case for a team version of the well-known game of matching pennies. The game can be shortly described as such: "*coordinate with your teammates to play a game of matching pennies against the opposing team, coordinate not and pay a penalty*". The penalty is set to $\frac{1}{2}$. For the interested reader, we defer to appendix C.2 the precise description of the game in a contracted tensor/table. Since every player has simply two actions, their probability vector can be represented by a single variable in $[0, 1]$. Considering the minimizing team **x** its players are $x_1, x_2$, while the players of the maximizing team **y** are $y_1, y_2$. The multiplayer of matching pennies is described by the utility function:

$$u(x_1, x_2, y_1, y_2) = -x_1x_2 - x_1y_1 - x_1y_2 + 1.5(x_1 + x_2) - x_2y_1 - x_2y_2 + y_1y_2 + 0.5(y_1 + y_2) - 1 \quad (2.6)$$

As we can see in Figures 2 and 3, multiplayer matching pennies game consists an excellent benchmark where all traditional gradient flow discretizations fail under perfect competition setting. Interestingly, we are not aware of a similar example in min-max literature and it has been our starting point for seeking new optimization techniques inspired by Control theory. Indeed, KPV-GDA variation with $(K, P) = (-1.1\mathbf{I}, 0.3\mathbf{I})$ achieves to converge to the unique mixed Nash Equilibrium of the game. In the following sections, we provide theorems that explained formally this long-run behavior of the examined dynamics.

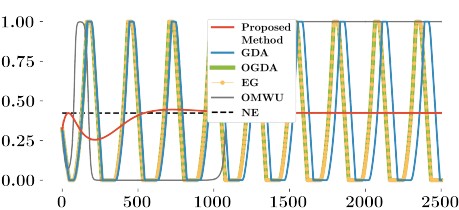
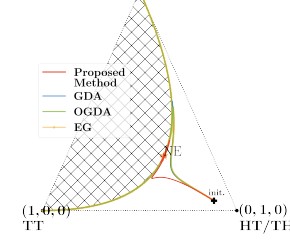

Figure 2: Multiplayer matching pennies under different algorithms

Figure 3: Projected Trajectory of Team A under different algorithms

## 3 OUR MAIN RESULTS

### 3.1 ON THE COMPLEXITY OF TWO-TEAM ZERO-SUM GAMES

We start this section by showing that computing a Nash equilibrium in two-team zero-sum games is computationally hard and thus getting a polynomial-time algorithm that computes a Nash equilibrium is unlikely.

**Theorem 3.1** (CLS-hard). *Computing a Nash equilibrium in two-team zero-sum games is* CLS-*hard.*

The main idea of the proof of Theorem 3.1 relies on a reduction of approximating Nash equilibria in congestion games, which has been shown to be complete for the interesting class of CLS, which contains the problem of continuous optimization. For concision, we defer the proof of the above theorem to the paper's supplement.

### 3.2 FIRST-ORDER METHODS FAIL TO STABILIZE

The negative computational complexity result we proved for two-team zero-sum games (Theorem 3.1) does not conclude the prospect of having algorithms (learning dynamics, first-order methods) that converge to Nash equilibria and thus can approximate them well enough. Unfortunately, we can even prove negative results about convergence to Nash equilibria in two-team zero-sum games of well-established methods broadly used in classic two-player zero-sum games.

In this section, we are going to construct a family of two-team zero-sum games with the property that GDA, OGDA, EG, and OMWU fail to stabilize to Nash equilibria. This result indicates how challenging and rich the setting of team zero-sum games can be and why provable guarantees about

convergence have not been established yet. Before defining the family of two-team zero-sum games, we prove an important theorem which states that GDA does not stabilize around mixed Nash equilibria. This fact is a stepping stone in constructing the family of team-zero sum games later. We present the proof of all of the below statements in detail in the paper's appendix.

**Weakly-stable Nash equilibrium (Kleinberg et al., 2009; Mehta et al., 2015).** Consider the set of Nash equilibria with the property that if any single randomizing agent of one team is forced to play any strategy in her current support with probability one, all other agents of the same team must remain indifferent between the strategies in their support. This type of Nash equilibria is called weakly-stable. Note that trivially pure Nash equilibria are weakly-stable. It has been shown that mixed Nash equilibria are not weakly-stable in generic games[5] (Kleinberg et al., 2009). We can show that Nash equilibria that are not weakly-stable Nash are actually unstable for GDA moreover, through standard dynamical systems machinery, that the set of initial conditions that converges to Nash equilibria that are not weakly-stable should be of measure zero. Formally, we prove that:

**Theorem 3.2** (Non weakly-stable Nash are unstable). *Consider a two-team zero-sum game with utility function of Team B ($\mathbf{y}$ vector) being $U(\mathbf{x}, \mathbf{y})$ and Team A ($\mathbf{x}$ vector) being $-U(\mathbf{x}, \mathbf{y})$. Moreover, assume that $(\mathbf{x}^*, \mathbf{y}^*)$ is a Nash equilibrium of full support that is not weakly-stable. It follows that the set of initial conditions so that GDA converges to $(\mathbf{x}^*, \mathbf{y}^*)$ is of measure zero for step size $\eta < \frac{1}{L}$ where $L$ is the Lipschitz constant of $\nabla U$.*

### 3.3 GENERALIZED MATCHING PENNIES (GMP)

Inspired by Theorem 3.2, in this section we construct a family of team zero-sum games so that GDA, OGDA, and EG methods fail to converge (if the initialization is a random point in the simplex, the probability of convergence of the aforementioned methods is zero). The intuition is to construct a family of games, each of which has only mixed Nash equilibria (that are not weakly-stable), i.e., the constructed games should lack pure Nash equilibria; using Theorem 3.2, it would immediately imply our claim for GDA. It turns out that OGDA and EG also fail to converge for the same family.

**Definition of GMP.** Consider a setting with two teams (Team $A$, Team $B$), each of which has $n = 2$ players. Inspired by the standard matching pennies game and the game defined in Schulman & Vazirani (2019a), we allow each agent $i$ to have two strategies/actions that is $S = \{H, T\}$ for both teams with $2^4$ possible strategy profiles. In case all the members of a Team choose the same strategy say $H$ or $T$ then the Team "agrees" to play $H$ or $T$ (otherwise the Team "does not agree").

Thus, in the case both teams "agree", the payoff of each team is actually the payoffs for the two-player matching pennies. If one team "agrees" and the other does not, the team that "agrees" gets payoff $\omega \in (0, 1)$ and the other team gets penalty $\omega$. If both teams fail to "agree", both teams get payoff zero. Let $x_i$ with $i \in \{1, 2\}$ be the probability that agent $i$ of Team $A$ chooses $H$ and $1 - x_i$ the probability that she chooses $T$. We also denote $\mathbf{x}$ the vector of probabilities for Team $A$. Similarly, we denote $y_i$ for $i \in \{1, 2\}$ be the probability that agent $i$ of Team $B$ chooses $H$ and $1 - y_i$ the probability that she chooses $T$ and $\mathbf{y}$ the probability vector.

The first fact about the game that we defined is that for $\omega \in (0, 1)$, there is only one Nash equilibrium $(\mathbf{x}^*, \mathbf{y}^*)$, which is the uniform, i.e., $x_1^* = x_2^* = y_1^* = y_2^* = \frac{1}{2}$ for all agents $i$.

**Lemma 3.3** (GMP has a unique Nash). *The Generalized Matching Pennies game exhibits a unique Nash equilibrium which is $(\mathbf{x}^*, \mathbf{y}^*) = ((\frac{1}{2}, \frac{1}{2}), (\frac{1}{2}, \frac{1}{2}))$.*

*Remark* 3.4. The fact that the game we defined has a unique Nash equilibrium that is in the interior of $[0, 1]^4$ is really crucial for our negative convergence results later in the section as we will show that it is not weakly-stable Nash equilibrium and the negative result about GDA will be a corollary due to Theorem 3.2. Please also note that if $\omega = 1$ then there are more Nash equilibria, in particular the $(\mathbf{0}, \mathbf{0}), (\mathbf{1}, \mathbf{0}), (\mathbf{0}, \mathbf{1}), (\mathbf{1}, \mathbf{1})$ are also Nash equilibria (which are pure).

The following Theorem is the main (negative) result of this section.

**Theorem 3.5** (GDA, OGDA, EG, and OMWU fail). *Consider GMP game with $\omega \in (0, 1)$. Assume that $\eta_{GDA} < \frac{1}{4}$, $\eta_{OGDA} < \min(\omega, \frac{1}{8})$, $\eta_{EG} < \frac{\omega}{2}$, and $\eta_{OMWU} < \min\left(\frac{1}{4}, \frac{\omega}{2}\right)$ (bound on the stepsize*

---

[5]Roughly speaking, games in which we add small Gaussian noise on every payoff.

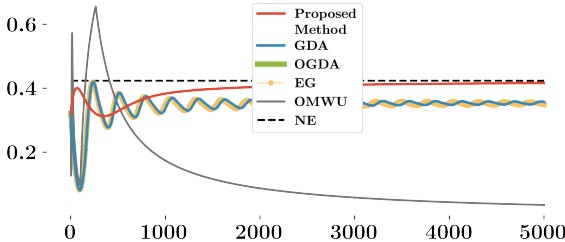

Figure 4: GDA, OGDA, & EG fail to converge to a Nash Equilibrium even in average

*for GDA, OGDA, EG, and OMWU methods respectively). It holds that the set of initial conditions so that GDA, OGDA, EG, OMWU converge (stabilize to any point) is of measure zero.*

*Remark* 3.6 (Average iterate also fails). One might ask what happens when we consider average iterates instead of the last iterate. It is well-known fact Syrgkanis et al. (2015) that the average iterate of no-regret algorithms converges to coarse correlated equilibria (CCE) so we expect that the average iterate stabilizes. Nevertheless, CCE might not be Nash equilibria. Indeed we can construct examples in which the average iterate of GDA, OGDA, and EG experimentally fail to stabilize to Nash equilibria. In particular, we consider a slight modification of GMP; players and strategies are the same but the payoff matrix has changed and can be found below (table on the right):

|  | $HH$ | $HT/TH$ | $TT$ |
|---|---|---|---|
| $HH$ | $1, -1$ | $\omega, -\omega$ | $-1, 1$ |
| $HT/TH$ | $-\omega, \omega$ | $0, 0$ | $-\omega, \omega$ |
| $TT$ | $-1, 1$ | $\omega, -\omega$ | $1, -1$ |

|  | $HH$ | $HT/TH$ | $TT$ |
|---|---|---|---|
| $HH$ | $2, -2$ | $\frac{1}{2}, -\frac{1}{2}$ | $-2, 2$ |
| $HT/TH$ | $-\frac{1}{2}, \frac{1}{2}$ | $0, 0$ | $-\frac{1}{2}, \frac{1}{2}$ |
| $TT$ | $-1, 1$ | $\frac{1}{2}, -\frac{1}{2}$ | $1, -1$ |

Table 1: GMP configurations of Theorem 3.5 (left) and Remark 3.6 (right)

Figure 4 illustrates that the average iterates of GDA, OGDA, and EG stabilize to points that are not Nash equilibria. Note that since our method (see next subsection) converges locally, the average iterate should converge locally to a Nash equilibrium.

## 3.4 WASH-OUT FILTERS & ADAPTIVE CONTROL

The aforementioned results indicate that to answer the tantalizing question of finding NE in two-team zero-sum games, our machinery should be broadened outside the limits of textbook optimization arsenal. The mainstay of this effort and our positive result is KPV-GDA method defined in (2.2), inspired by the adaptive control toolbox and washout filters. Our main statement shows that KPV-GDA stabilizes around any Nash equilibrium for appropriate choices of matrices $K, P$. The formal theorem is given below:

**Theorem 3.7** (KPV-GDA stabilizes). *Consider a team zero-sum game so that the utility of Team B is $U(\mathbf{x}, \mathbf{y})$ and hence the utility of Team A is $-U(\mathbf{x}, \mathbf{y})$ and a Nash equilibrium $(\mathbf{x}^*, \mathbf{y}^*)$ of the game. Moreover we assume*

$$\begin{pmatrix} -\nabla^2_{\mathbf{xx}}U(\mathbf{x}^*, \mathbf{y}^*) & -\nabla^2_{\mathbf{xy}}U(\mathbf{x}^*, \mathbf{y}^*) \\ \nabla^2_{\mathbf{yx}}U(\mathbf{x}^*, \mathbf{y}^*) & \nabla^2_{\mathbf{yy}}U(\mathbf{x}^*, \mathbf{y}^*) \end{pmatrix} \text{ is invertible.}$$

*For any fixed step size $\eta > 0$, we can always find matrices $K, P$ so that KPV-GDA method defined in (2.2) converges locally to $(\mathbf{x}^*, \mathbf{y}^*)$.*

The latter statement concerns the existence of matrices $K, P$. Below, we provide a sufficient condition under which a simple parametrization of $K, P$ (provably) guarantees convergence.

**Theorem 3.8.** *Consider a two-team zero-sum game so that the utility of Team B is $U(\mathbf{x}, \mathbf{y})$ and hence the utility of Team A is $-U(\mathbf{x}, \mathbf{y})$ and a Nash equilibrium $(\mathbf{x}^*, \mathbf{y}^*)$ of the game. Moreover let*

$$H := \begin{pmatrix} -\nabla^2_{\mathbf{xx}} U(\mathbf{x}^*, \mathbf{y}^*) & -\nabla^2_{\mathbf{xy}} U(\mathbf{x}^*, \mathbf{y}^*) \\ \nabla^2_{\mathbf{yx}} U(\mathbf{x}^*, \mathbf{y}^*) & \nabla^2_{\mathbf{yy}} U(\mathbf{x}^*, \mathbf{y}^*) \end{pmatrix}.$$

*and $E$ be the set of eigenvalues $\rho$ of $H$ with real part positive, that is $E = \{H's \text{ eigenvalues } \rho : Re(\rho) > 0\}$. We assume that $H$ is invertible and moreover*

$$\beta = \min_{\rho \in E} \frac{Re(\rho)^2 + Im(\rho)^2}{Re(\rho)} > \max_{\rho \in E} Re(\rho) = \alpha. \tag{3.1}$$

*We set $K = k \cdot \mathbf{I}$, $P = p \cdot \mathbf{I}$. There exist small enough step size $\eta > 0$ and scalar $p > 0$ and for any $k \in (-\beta, -\alpha)$ so that KPV-GDA method defined in (2.2) with chosen $K, P$ converges locally to $(\mathbf{x}^*, \mathbf{y}^*)$.*

## 4 EXPERIMENTS

In this section, we perform a series of numerical experiments to validate our theoretical findings. Our experiment setting includes a 2-D Gaussian Mixture Model with 8 modes. Our architecture includes 8 "shallow" generators and discriminators with 2 layers of 2-16-2 ReLUs activations, compared with a giant single-agent GAN with 4 layers of 2-128-256-1024-2 activations. Interestingly, the giant one fails in a double sense; It demonstrates both mode-collapsing and mode-drop phenomena without stabilizing. On the other hand, our architecture with a small number of neurons achieves to fit the data well. We defer the discussion of the experiments with multi-generators multi-discriminator architectures for CIFAR-10 again to the paper's supplement.

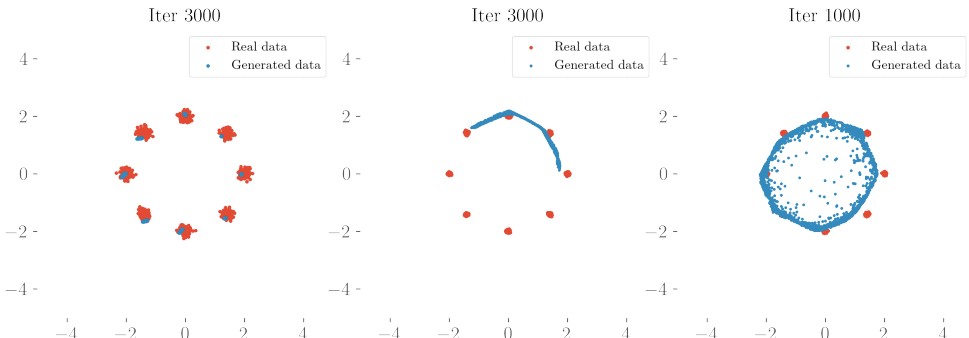

(a) Each generator of MGAN learns one mode of 8-GMM  (b) Mode Collapse of single-agent GANs  (c) Single-agent GAN can't discriminate between the modes

## 5 CONCLUSIONS AND OPEN PROBLEMS

In this paper, we have presented a number of negative results about the problem of finding a Nash equilibrium in team zero-sum games and moreover about the inability of commonly used methods for min-max optimization such as GDA, OGDA, and EG to stabilize. We also presented a method (called KPV-GDA) that manages to stabilize around Nash equilibria. Given these results, a number of interesting open questions emerge.

**Open Questions.** One question for future consideration is the global convergence and the rates of convergence of KPV-GDA method. We believe that the KPV-GDA converges globally for an appropriate choice of matrices $K, P$. One other possible direction is to find a systematic way to get the matrices $K, P$.

REPRODUCIBILITY STATEMENT

In our submission folder, we provide all the necessary additional technical materials and complete proofs of the main draft's statements in the appendix section. We also uploaded the code of our experiments (Python/PyTorch/Tensorflow).

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

# A APPENDIX

## A.1 FURTHER RELATED WORK

The properties of team games have given rise to a vast corpus of literature which we cannot hope to review here; for an appetizer, we refer the reader to Kim et al. (2019); Gold (2005) and reference therein. On the other hand, the convergence of first-order methods to NE, like GDA, OGDA, or Extragradient in *team games* (even finite ones) is nowhere near as well understood.

A notable exception to this is the case of *team-maxmin equilibria* (TME), a notion introduced by von Stengel & Koller (1997). In this category of two-team zero-sum games, a team of $n-$players plays defensively against a single *adversarial* player. TMEs provide crucial information about the robustness since they give the team players the highest payoff they employ uncoordinated strategies. Unfortunately, TMEs correspond to the solution of a non–linear non–convex program. (Basilico et al., 2017b) provided heuristics with iterative LP and support enumeration methods with proved approximation guarantees. For the case of extensive or large-sized normal form games, Basilico et al. (2017a); Celli & Gatti (2018) introduced a relaxed notion of *team-maxmin equilibrium with coordination device* (TMECor). In TMECor solutions, the team of players shares their strategies *ex-ante*[6] against the adversary. While TMECor shares some properties of NEs in zero-sum two-player games (e.g., exchangeability), the proposed algorithms by (Celli & Gatti, 2018) leverages tools like the Mixed-Integer Linear Program (MILP) that involves a large number of integer variables. To tackle heuristically these hurdles, it (Zhang et al., 2020; Zhang & An, 2020) had been proposed modified versions of incremental strategy generation (ISG) algorithm (See (McMahan et al., 2003)) for large-sized games.

Describing the history and the literature surrounding multi-agent GANs would take us too far afield, so we do not attempt it. Scratching only the surface, we mention that: *a)* In MGAN (Hoang et al., 2018) and MAD-GAN (Li et al., 2019), we have a mixture of generative models with asymptotically infinite capacity networks. *b)* In the seminal case of MIX+GAN (Arora et al., 2017), an extra regularization term is typically added to discourage the weights of a mixture of generators being too far away from uniform. *c)* In Stackelberg GAN, (Zhang et al., 2018) exploit the leader-followers relation of discriminator vs generators to achieve smaller minimax gap and better Frechét metrics. For the case of *multiple-discriminators*: *d)* In GMAN (Durugkar et al., 2016), an increased number of agents are used to achieve higher quality images, while *e)* MD-GAN (Hardy et al., 2019) is designed to generalize over different datasets and multiple tasks that are spread on multiple workers.

Interestingly, in the continuous regime, gradient flow has been recently studied under the perspective of dynamical systems. In the setting of *Hidden Games*, two competitive non-convex operators join in a two-player min-max game. In Piliouras & Schulman (2018); Vlatakis-Gkaragkounis et al. (2019), the authors exploit the bi-linear form of the hidden game to prove antithetical to convergence, like spurious fixed points, cycles, and generalized high-dimensional recurrence phenomena. Additionally, in the case of general convex-concave settings, Flokas et al. (2021) proved that local stability can be achieved around the von-Neumann equilibria while a global convergent result can be described for strictly convex hidden games. A crucial comparison of the hidden model with the *team games* as described in our setting is the capacity & independence of the player's choice.

While the main contribution of this work is to highlight the inherent difficulties that the above heuristic architectures are cursed, our theoretical findings in team competition hint also at new optimization schemes like KPV-GDA whose performance is illustrated in Section 4 with delicate network architectures that contain even multiple generators and discriminators simultaneously.

## A.2 DISCUSSION ABOUT DIFFERENT SOLUTION CONCEPTS

In this subsection, we would like to stress some differences between the different equilibrium notions ( *TMECor, TME, and per-player NE* ) explaining the game-theoretic rationalism of our choice.

Initially, it is easy to see that TMECor similarly with the Coarse correlated equilibria can achieve better social welfare but as in the case of congestion games, found in literature, their price of anarchy

---

[6]That is, the team members are allowed to discuss and agree on tactics before the game starts, but they cannot communicate during the game.

can become significantly worse Roughgarden (2009). In order to give a better result, TMECor requests ana priori knowledge of the game. For example, as Zhang et al. (2020) mentioned: *For example, in multiplayer poker games, a team may play against an adversary player, but they cannot communicate and discuss their strategy during the game due to the rule.*

However in many nowadays challenging AI models, the agents/players of each team do not necessarily have knowledge of the actual game. Thus, the team players can only decide in advance only about their own dynamics. Thus, the main conceptual reason that we focus on NE per player in comparison with the aforementioned TME relaxations is to study the interplay between the individual and team incentives in competitive tasks. It is also a way of reasoning about games in which players do not necessarily need to know in which team they fall into in advance.

Problems of this kind are prevalent in modern ML applications like strategic conflict resolution Leibo et al. (2017), coordination between autonomous vehicles Cao et al. (2012) , or collaboration of agents in defensive escort teams Sheikh et al. (2019). In this kind of games each agent is simultaneously working towards maximizing its own payoff *(local reward)* as well as the collective success of the team *(global reward)*.

As it is stated again in Zhang et al. (2020), *Celli and Gatti (2018) show that the "ex ante" coordination can be modeled using a coordination device, assuming that the adversary does not observe any signal from the device. The team members agree on a planned strategy (e.g., a mixed strategy) in the planning phase, and then, just before the game starts, the coordination device randomly picks a pure joint strategy (from the planned strategy) for the team members to act upon.*

Therefore, this TMECor interpretation would critically correspond to a high-level two-metaplayer zero-sum game, where the metaplayer is the team as a whole entity with every individual player in perfect coordination with its teammates.

Finally, an additional reason, the notion of "per player Nash equilibria" can be seen as a smoother figure of merit for the performance of a defensive team against an adversary. Notably, in the vast majority of the aforementioned literature, a single superpowerful adversary has been used. However, using a single-agent adversary can describe a universally fair model.By contrast, the notion of per player NE expresses the model hurdles of both simultaneous *team* intra-collaboration and inter-competition.

## B  GAME DYNAMICS

The following algorithms are ubiquitous in the literature of $\min \max$ optimization and have known variable success under different settings. Gradient descent-ascent (GDA) is the prototypical method upon which Optimistic Gradient Descent (OGDA), Extra Gradient (EG), and our proposed method $K, P$-vaned Gradient Descent-Ascent (KPV-GDA) are built. Optimistic Multiplicative Weights Update Method is the optimistic variant of the Multiplicative Weights Update Method.

### B.1  GRADIENT DESCENT-ASCENT

$$\begin{cases} x_i^{(k+1)} = \Pi_{\mathcal{X}_i}\left\{ x_i^{(k)} - \eta \nabla_{x_i} f(\mathbf{x}^{(k)}, \mathbf{y}^{(k)}) \right\} \\ y_j^{(k+1)} = \Pi_{\mathcal{Y}_j}\left\{ y_j^{(k)} + \eta \nabla_{y_j} f(\mathbf{x}^{(k)}, \mathbf{y}^{(k)}) \right\} \end{cases}$$

### B.2  OPTIMISTIC GRADIENT DESCENT-ASCENT

$$\begin{cases} x_i^{(k+1)} = \Pi_{\mathcal{X}_i}\left\{ x_i^{(k)} - 2\eta \nabla_{x_i} f(\mathbf{x}^{(k)}, \mathbf{y}^{(k)}) + \eta \nabla_{x_i} f(\mathbf{x}^{(k-1)}, \mathbf{y}^{(k-1)}) \right\} \\ y_j^{(k+1)} = \Pi_{\mathcal{Y}_j}\left\{ y_j^{(k)} + 2\eta \nabla_{y_j} f(\mathbf{x}^{(k)}, \mathbf{y}^{(k)}) - \eta \nabla_{y_j} f(\mathbf{x}^{(k-1)}, \mathbf{y}^{(k-1)}) \right\} \end{cases}$$

### B.3  EXTRA GRADIENT METHOD

$$\begin{cases} x_i^{(k+\frac{1}{2})} = \Pi_{\mathcal{X}_i}\left\{ x_i^{(k)} - \eta \nabla_{x_i} f(\mathbf{x}^{(k)}, \mathbf{y}^{(k)}) \right\}, & x_i^{(k+1)} = \Pi_{\mathcal{X}_i}\left\{ x_i^{(k)} - \eta \nabla_{x_i} f(\mathbf{x}^{(k+\frac{1}{2})}, \mathbf{y}^{(k+\frac{1}{2})}) \right\} \\ y_j^{(k+\frac{1}{2})} = \Pi_{\mathcal{Y}_j}\left\{ y_j^{(k)} + \eta \nabla_{y_j} f(\mathbf{x}^{(k)}, \mathbf{y}^{(k)}) \right\}, & y_j^{(k+1)} = \Pi_{\mathcal{Y}_j}\left\{ y_j^{(k)} + \eta \nabla_{y_j} f(\mathbf{x}^{(k+\frac{1}{2})}, \mathbf{y}^{(k+\frac{1}{2})}) \right\} \end{cases}$$

### B.4 OPTIMISTIC MULTIPLICATIVE WEIGHTS UPDATE METHOD

$$
\begin{cases}
x_i^{(k+1)} = x_i^{(k)} \dfrac{\exp\left(-2\eta\nabla_{x_i} f(\mathbf{x}^{(k)}, \mathbf{y}^{(k)}) + \eta\nabla_{x_i} f(\mathbf{x}^{(k-1)}, \mathbf{y}^{(k-1)})\right)}{\sum_j x_j^{(k)} \exp\left(-2\eta\nabla_{x_i} f(\mathbf{x}^{(k)}, \mathbf{y}^{(k)}) + \eta\nabla_{x_i} f(\mathbf{x}^{(k-1)}, \mathbf{y}^{(k-1)})\right)} \\[2em]
y_j^{(k+1)} = y_j^{(k)} \dfrac{\exp\left(2\eta\nabla_{y_j} f(\mathbf{x}^{(k)}, \mathbf{y}^{(k)}) - \eta\nabla_{y_j} f(\mathbf{x}^{(k-1)}, \mathbf{y}^{(k-1)})\right)}{\sum_j y_j^{(k)} \exp\left(2\eta\nabla_{y_j} f(\mathbf{x}^{(k)}, \mathbf{y}^{(k)}) - \eta\nabla_{y_j} f(\mathbf{x}^{(k-1)}, \mathbf{y}^{(k-1)})\right)}
\end{cases}
$$

### B.5 $K, P$-VANED GRADIENT DESCENT-ASCENT

$$
\begin{cases}
\mathbf{z}^{(k+1)} = \Pi_{\mathcal{Z}}\left\{\mathbf{z}^{(k)} + \eta\binom{-\nabla_{\mathbf{x}} f(\mathbf{z}^{(k)})}{\nabla_{\mathbf{y}} f(\mathbf{z}^{(k)})} + \eta K(\mathbf{z}^{(k)} - \boldsymbol{\theta}^{(k)})\right\} \\[1em]
\boldsymbol{\theta}^{(k+1)} = \Pi_{\mathcal{Z}}\left\{\boldsymbol{\theta}^{(k)} + \eta P(\mathbf{z}^{(k)} - \boldsymbol{\theta}^{(k)})\right\}
\end{cases}
$$

Notations $x_i^{(k)}$ ( or $y_j^{(k)}$) stand for the strategy vector of the $i$-th minimizing (or $j$-th maximizing) agent at time-step $k$. The step size is denoted as $\eta$. Operators $\Pi_{\mathcal{X}}, \Pi_{\mathcal{Y}}, \Pi_{\mathcal{Z}}$ are the projection operators to the corresponding simplices. The projection operator is important since the context of operation is that of constrained optimization and remaining inside the set of feasible solutions is not guaranteed at every step except for the case of the OMWU method.

## C DERIVATION OF THE MIN-MAX OBJECTIVE IN EQUATION 2.5

By the definition of Variance, we get that : $\text{Var}[x_i] = \mathbb{E}[x_i^2] - (\mathbb{E}[x_i])^2 \Leftrightarrow \mathbb{E}[x_i^2] = \text{Var}[x_i] + (\mathbb{E}[x_i])^2$. More precisely for any $\mathbf{x} \sim \mathcal{N}(\mu, I)$, we get $\mathbb{E}[x_i^2] = 1 + \mu_i^2$.

Additionally, after calculations, we can get that:

$$\mathbb{E}_{z\sim\mathcal{N}(0,\boldsymbol{I})}[(z_i + \theta_i)^2] = \text{Var}[z_i + \theta_i] + (\mathbb{E}[z_i + \theta_i])^2 \xRightarrow{\theta_i \text{ const.}}$$

$$\mathbb{E}_{z\sim\mathcal{N}(0,\boldsymbol{I})}[(z_i + \theta_i)^2] = \text{Var}[z_i] + (\mathbb{E}[z_i] + \mathbb{E}[\theta_i])^2 \Rightarrow$$

$$\mathbb{E}_{z\sim\mathcal{N}(0,\boldsymbol{I})}[(z_i + \theta_i)^2] = \text{Var}[z_i] + (0 + \theta_i)^2 \Rightarrow$$

$$\mathbb{E}_{z\sim\mathcal{N}(0,\boldsymbol{I})}[(z_i + \theta_i)^2] = 1 + \theta_i^2$$

Moreover, it is easy to check that for a mixture $\mathcal{D}(\mu_1, \sigma_1\boldsymbol{I}, \mu_2, \sigma_2\boldsymbol{I}, \pi_1, \pi_2)$ of two multi-dimensional normal distributions $\mathcal{N}(\mu_1, \sigma_1\boldsymbol{I}), \mathcal{N}(\mu_1, \sigma_1\boldsymbol{I})$ with corresponding weights $\pi_1, \pi_2$:

$$\text{Var}_{x\sim\mathcal{D}}[x_i] = \pi_1\sigma_1^2 + \pi_2\sigma_2^2 + \pi_1\mu_{1,i}^2 + \pi_2\mu_{2,i}^2 - (\pi_1\mu_{1,i} + \pi_2\mu_{2,i})^2$$

Specifically for the case of $\mu_1 = \mu = -\mu_2$, this equivalent with:

$$\text{Var}[x_i] = 1 + \mu_i^2 - (\pi_1\mu_i - \pi_2\mu_i)^2$$

Hence:

$$\mathbb{E}[x_i^2] = \text{Var}[x_i] + (\pi_1\mu_i - \pi_2\mu_i)^2 = 1 + \mu_i^2$$

So we can apply the following manipulations in the objective (The notation $[x_i^2]$ stands for the vector that has as entries the values of the vector $\mathbf{x}$ squared):

$$
\begin{aligned}
\max_{\mathbf{v},\mathbf{w}} \min_{\boldsymbol{\theta},p} \; & \mathbb{E}_{\mathbf{y}\sim real}\Big[D_{\mathbf{v}}(\mathbf{y}) + D_{\mathbf{w}}(\mathbf{y})\Big] - \\
& - \mathbb{E}_{z\sim\mathcal{N}(0,\boldsymbol{I}),\zeta\sim\mathcal{N}(0,1)}\Big[G_p(\zeta) \cdot \big(D_{\mathbf{v}}(-G_{\boldsymbol{\theta}}(\mathbf{y})) + D_{\mathbf{w}}(G_{\boldsymbol{\theta}}(\mathbf{y}))\big) + \\
& + (1 - G_p(\zeta)) \cdot \big(D_{\mathbf{v}}(G_{\boldsymbol{\theta}}(\mathbf{y})) + D_{\mathbf{w}}(G_{\boldsymbol{\theta}}(\mathbf{y}))\big)\Big] =
\end{aligned}
$$

$$\max_{\mathbf{v},\mathbf{w}} \min_{\boldsymbol{\theta},p} \ \mathbb{E}_{\mathbf{y}\sim real}\Big[\langle\mathbf{v},\mathbf{y}\rangle + \langle\mathbf{w},[y_i^2]\rangle\Big] -$$

$$-\mathbb{E}_{z\sim\mathcal{N}(0,\mathbf{I}),\zeta\sim\mathcal{N}(0,1)}\Big[(p+\zeta)\cdot\Big(\langle\mathbf{v},-(\mathbf{z}+\boldsymbol{\theta})\rangle + \langle\mathbf{w},[(\theta_i+z_i)^2]\rangle\Big) +$$

$$+ \big(1-(p+\zeta)\big)\cdot\Big(\langle\mathbf{v},-(\mathbf{z}+\boldsymbol{\theta})\rangle + \langle\mathbf{w},[(-(\theta_i+z_i))^2]\rangle\Big)\Big] =$$

$$\max_{\mathbf{v},\mathbf{w}} \min_{\boldsymbol{\theta},p} \ \mathbb{E}_{\mathbf{y}\sim real}\Big[\langle\mathbf{v},\mathbf{y}\rangle\Big] + \mathbb{E}_{\mathbf{y}\sim real}\Big[\langle\mathbf{w},[y_i^2]\rangle\Big] -$$

$$-\mathbb{E}_{z\sim\mathcal{N}(0,\mathbf{I}),\zeta\sim\mathcal{N}(0,1)}\Big[(p+\zeta)\cdot\Big(\langle\mathbf{v},\mathbf{z}+\boldsymbol{\theta}\rangle + \langle\mathbf{w},[(\theta_i+z_i)^2]\rangle\Big) +$$

$$+ \big(1-(p+\zeta)\big)\cdot\Big(\langle\mathbf{v},-(\mathbf{z}+\boldsymbol{\theta})\rangle + \langle\mathbf{w},[(\theta_i+z_i)^2]\rangle\Big)\Big] =$$

$$\max_{\mathbf{v},\mathbf{w}} \min_{\boldsymbol{\theta},p} \ \Big\langle\mathbf{v},\mathbb{E}_{\mathbf{y}\sim real}\big[\mathbf{y}\big]\Big\rangle + \Big\langle\mathbf{w},\mathbb{E}_{\mathbf{y}\sim real}\big[[y_i^2]\big]\Big\rangle -$$

$$-p\cdot\Big(\Big\langle\mathbf{v},\mathbb{E}_{z\sim\mathcal{N}(0,\mathbf{I})}\big[(\mathbf{z}+\boldsymbol{\theta})\big]\Big\rangle + \Big\langle\mathbf{w},\mathbb{E}_{z\sim\mathcal{N}(0,\mathbf{I})}\big[[(\theta_i+z_i)^2]\big]\Big\rangle\Big) -$$

$$-\big(1-p\big)\Big(\cdot\Big\langle\mathbf{v},\mathbb{E}_{z\sim\mathcal{N}(0,\mathbf{I})}\big[-(\mathbf{z}+\boldsymbol{\theta})\big]\Big\rangle + \Big\langle\mathbf{w},\mathbb{E}_{z\sim\mathcal{N}(0,\mathbf{I})}\big[[(\theta_i+z_i)^2]\big]\Big\rangle\Big) =$$

$$\max_{\mathbf{v},\mathbf{w}} \min_{\boldsymbol{\theta},p} \ \Big\langle\mathbf{v},\big(\pi_1\boldsymbol{\mu}+\pi_2(-\boldsymbol{\mu})\big)\Big\rangle + \Big\langle\mathbf{w},[1+\mu_i^2]\Big\rangle -$$

$$-p\cdot\Big(\langle\mathbf{v},\boldsymbol{\theta}\rangle + \Big\langle\mathbf{w},[(\theta_i^2+1)]\Big\rangle\Big) - (1-p)\cdot\Big(\langle\mathbf{v},-\boldsymbol{\theta}\rangle + \Big\langle\mathbf{w},[(\theta_i^2+1)]\Big\rangle\Big) =$$

$$\max_{\mathbf{v},\mathbf{w}} \min_{\boldsymbol{\theta},p} \ (\pi_1-\pi_2)\mathbf{v}^T\boldsymbol{\mu} - 2p\mathbf{v}^T\boldsymbol{\theta} + \mathbf{v}^T\boldsymbol{\theta} + \sum_i w_i(\mu_i^2-\theta_i^2)$$

## C.1 Proof of Theorem 3.1

We will reduce the problem of finding a Nash equilibrium in congestion games to the problem of finding a Nash equilibrium in two-team zero-sum games. The result then will follow since computing Nash equilibria in congestion games is CLS-hard (Babichenko & Rubinstein (2021)).

As a recent result (Fearnley et al., 2021) shows CLS is equal to the intersection to PLS and PPAD, two important classes of total problems. PPAD captures diverse problems in combinatorics and (non-)cooperative game theory, like the $\varepsilon$-approximation of a mixed Nash Equilibrium in a graphical game or the computation of market equilibria. PLS, for "Polynomial Local Search", captures problems of finding a local minimum of an objective function f, in contexts where any candidate solution x has a local neighbourhood within which we can readily check for the existence of some other point having a lower value of f. Many diverse local optimization problems have been shown complete for PLS, attesting to its importance. Examples include searching for a local optimum of the TSP according to the Lin-Kernighan heuristic (Papadimitriou, 1992), and finding pure Nash equilibria in many-player congestion games (Fabrikant et al., 2004). The complexity class CLS ("Continuous Local Search") was introduced by Daskalakis and Papadimitriou (Daskalakis & Papadimitriou, 2011) to classify various important problems that lie in both PPAD and PLS. CLS is seen as a strong candidate for capturing the complexity of some of those important problems, like the general versions of Banach's fixed point theorem, computation of KKT points, computation of gradient descent fixed points etc.

In our reduction, a *congestion game* is defined by the tuple $(N; E; (S_i)_{i\in N}; (c_e)_{e\in E})$ where $N$ is the set of *agents*, $E$ is a set of *resources* (also known as *edges* or *facilities*), and each player $i$ has a set $S_i$ of subsets of $E$. Each strategy $s_i \in S_i$ is a set of edges (a *path*), and $c_e$ is a cost (negative utility) function associated with facility $e$. For a strategy profile $\mathbf{s} = (s_1, s_2, \ldots, s_N)$, the cost of player $i$ is given by $c_i(\mathbf{s}) = \sum_{e\in s_i} c_e(\ell_e(\mathbf{s}))$, where $\ell_e(\mathbf{s})$ is the number of players using $e$ in $\mathbf{s}$ (the load of edge $e$). It is a well-known result Rosenthal (1973) that congestion games exhibit a potential function $\Phi(\mathbf{s})$, that

$$\Phi(\mathbf{s}) = \sum_{e\in E}\sum_{j=1}^{\ell_e(\mathbf{s})} c_e(j)$$

with the property that if any agent $i$ changes her strategy to $s_i'$ it holds that

$$\Phi(s_i',\mathbf{s}_{-i}) - \Phi(s_i,\mathbf{s}_{-i}) = c_i(s_i',\mathbf{s}_{-i}) - c_i(s_i,\mathbf{s}_{-i}).$$

**Reduction.** Consider a congestion game $(N; E; (S_i)_{i \in N}; (c_e)_{e \in E})$ with $n = |N|$ players and potential function $\Phi$. We define a team zero-sum game as follows: Team $A$ has $n$ players, in which each agent $i$ chooses strategies from $S_i$. Team $B$ has $n$ players, with each agent $j$ having only one possible choice (singleton set of actions) call it $d$, i.e., these are dummy players. If players from Team $A$ choose strategy profile $\mathbf{s}$ (Team B has only one choice) then they get utility $u_A(\mathbf{s}, d) = -\Phi(s)$. The utility members of Team B get is $u_B(\mathbf{s}, d) = \Phi(s)$.

Let $\mathbf{x}^* \equiv (x_1^*, ..., x_n^*)$ and $(d, ..., d)$ be a (possibly mixed) Nash equilibrium in the team zero-sum game we defined. We shall show that $(x_1^*, ..., x_n^*)$ is a Nash equilibrium of the original congestion game and the reduction will be complete. Aiming for contradiction, suppose $(x_1^*, ..., x_n^*)$ is not a Nash equilibrium of the original congestion game. Then there exists an agent $i$ that can deviate from strategy $x_i^*$ to $\tilde{x}_i$ and decrease her expected cost. Hence we have that

$$0 < \mathbb{E}_{\mathbf{s} \sim \mathbf{x}^*}[c_i(\mathbf{s})] - \mathbb{E}_{\mathbf{s} \sim (\tilde{x}_i^*, \mathbf{x}_{-i}^*)}[c_i(\mathbf{s})]$$
$$= \mathbb{E}_{\mathbf{s} \sim \mathbf{x}^*}[\Phi(\mathbf{s})] - \mathbb{E}_{\mathbf{s} \sim (\tilde{x}_i^*, \mathbf{x}_{-i}^*)}[\Phi(\mathbf{s})] \quad \text{(Property of potential)}.$$

We conclude that $\mathbb{E}_{\mathbf{s} \sim \mathbf{x}^*}[u_A(\mathbf{s}, d)] = -\mathbb{E}_{\mathbf{s} \sim \mathbf{x}^*}[\Phi(\mathbf{s})] < -\mathbb{E}_{\mathbf{s} \sim (\tilde{x}_i^*, \mathbf{x}_{-i}^*)}[\Phi(\mathbf{s})] = \mathbb{E}_{\mathbf{s} \sim (\tilde{x}_i^*, \mathbf{x}_{-i}^*)}[u_A(\mathbf{s}, d)]$ which is a contradiction since $(x_1^*, ..., x_n^*)$ is a Nash equilibrium for the team zero-sum game hence if player $i$ deviates, her payoff (i.e., the payoff of her Team) should not increase.

## C.2 MULTIPLAYER MATCHING PENNIES

Below, we present the exact definition of the (2x2) two-team of two-players matching pennies that we discuss in Sec

|  |  | Team $B$ | | |
|---|---|---|---|---|
|  |  | $HH$ | $HT/TH$ | $TT$ |
| | $HH$ | $-1, 1$ | $-1/2, 1/2$ | $1, -1$ |
| Team $A$ | $HT/TH$ | $1/2, -1/2$ | $0, 0$ | $1/2, -1/2$ |
| | $TT$ | $1, -1$ | $-1/2, 1/2$ | $-1, 1$ |

## C.3 PROOF OF THEOREM 3.2

Since $(\mathbf{x}^*, \mathbf{y}^*)$ is not weakly-stable, there exist players $i, j$ from the same team (say $B$ without loss of generality) and strategies $k, l, l'$ so that if $i$ is forced to play $k$, then $j$'s best response is $l$ and that gives larger payoff than another strategy $l'$ in her support. Formally it holds that (by multi-linearity of $U$)

$$\text{(payoff if } i, j \text{ choose } k, l) \quad \frac{\partial^2 U(\mathbf{x}^*, \mathbf{y}^*)}{\partial y_{ik} \partial y_{jl}} > \frac{\partial^2 U(\mathbf{x}^*, \mathbf{y}^*)}{\partial y_{ik} \partial y_{jl'}} \quad \text{(payoff if } i, j \text{ choose } k, l'), \quad \text{(C.1)}$$

and also $\frac{\partial U(\mathbf{x}^*, \mathbf{y}^*)}{\partial y_{jl}} = \frac{\partial U(\mathbf{x}^*, \mathbf{y}^*)}{\partial y_{jl'}}$ (*). We shall show that $\nabla^2_{\mathbf{yy}} U(\mathbf{x}^*, \mathbf{y}^*)$ has a strictly positive eigenvalue tangent in the product of simplices (note that if we were working with $A$, we would show that $\nabla^2_{\mathbf{xx}} U(\mathbf{x}^*, \mathbf{y}^*)$ has a strictly negative eigenvalue). Consider a vector of size $\sum_i |S_i|$, (where $|S_i|$ is the cardinality of the strategy space of agent $i$ in Team $B$) which has 1 at coordinates $(i, k), (j, l)$, $-1$ at coordinate $(j, l')$ and from which we subtract $y_i^*$; we denote by $\mathbf{v}$ the resulting vector. We shall show that $\mathbf{v}^\top \nabla^2_{\mathbf{xx}} U(\mathbf{x}^*, \mathbf{y}^*) \mathbf{v} < 0$.

By multilinearity of $U$ it follows that $\frac{\partial^2 U}{\partial y_{js}\partial y_{js'}} = 0$ (**) for all agents $j$ and strategies $s, s'$ and the same is true for $x$ variables (team A). We conclude that

$$\frac{1}{2}\mathbf{v}^\top \nabla^2_{\mathbf{yy}} U(\mathbf{x}^*, \mathbf{y}^*)\mathbf{v} = \frac{\partial^2 U(\mathbf{x}^*, \mathbf{y}^*)}{\partial y_{ik}\partial y_{jl}} - \frac{\partial^2 U(\mathbf{x}^*, \mathbf{y}^*)}{\partial y_{ik}\partial y_{jl'}} - \sum_{s\in S_i} y^*_{is}\left(\frac{\partial^2 U(\mathbf{x}^*, \mathbf{y}^*)}{\partial y_{is}\partial y_{jl}} - \frac{\partial^2 U(\mathbf{x}^*, \mathbf{y}^*)}{\partial y_{is}\partial y_{jl'}}\right)$$

$$= \frac{\partial^2 U(\mathbf{x}^*, \mathbf{y}^*)}{\partial y_{ik}\partial y_{jl}} - \frac{\partial^2 U(\mathbf{x}^*, \mathbf{y}^*)}{\partial y_{ik}\partial y_{jl'}} - \left(\frac{\partial U(\mathbf{x}^*, \mathbf{y}^*)}{\partial y_{jl}} - \frac{\partial U(\mathbf{x}^*, \mathbf{y}^*)}{\partial y_{jl'}}\right)$$

$$\stackrel{(*)}{=} \frac{\partial^2 U(\mathbf{x}^*, \mathbf{y}^*)}{\partial y_{ik}\partial y_{jl}} - \frac{\partial^2 U(\mathbf{x}^*, \mathbf{y}^*)}{\partial y_{ik}\partial y_{jl'}} \stackrel{(C.1)}{>} 0.$$

Therefore $\nabla^2_{\mathbf{yy}} U(\mathbf{x}^*, \mathbf{y}^*)$ has a positive eigenvalue and as a result

$$R := \begin{pmatrix} -\nabla^2_{\mathbf{xx}} U(\mathbf{x}^*, \mathbf{y}^*) & \mathbf{0} \\ \mathbf{0} & \nabla^2_{\mathbf{yy}} U(\mathbf{x}^*, \mathbf{y}^*) \end{pmatrix} \tag{C.2}$$

must have a positive and a negative eigenvalue (since the trace is zero).

We consider the Jacobian of the GDA dynamics at $(\mathbf{x}^*, \mathbf{y}^*)$. The corresponding matrix is the following:

$$J_{\text{GDA}} = \mathbf{I} + \eta \begin{pmatrix} -\nabla^2_{\mathbf{xx}} U(\mathbf{x}^*, \mathbf{y}^*) & -\nabla^2_{\mathbf{xy}} U(\mathbf{x}^*, \mathbf{y}^*) \\ \nabla^2_{\mathbf{yx}} U(\mathbf{x}^*, \mathbf{y}^*) & \nabla^2_{\mathbf{yy}} U(\mathbf{x}^*, \mathbf{y}^*) \end{pmatrix}, \tag{C.3}$$

We will show that $J_{\text{GDA}}$ has an eigenvalue (possible complex) with an absolute value greater than one. It suffices to show that $J_{\text{GDA}} - \mathbf{I}$ has an eigenvalue with positive real part (because then $J_{\text{GDA}}$ would have an eigenvalue with real part greater than 1 and hence magnitude greater than one). Due to (**), we get that $J_{\text{GDA}} - \mathbf{I}$ has trace zero. To reach contradiction suppose that no eigenvalue of $J_{\text{GDA}} - \mathbf{I}$ has positive real part, then all eigenvalues of $J_{\text{GDA}} - \mathbf{I}$ should be imaginary or zero. But an imaginary eigenvalue (that is not zero) also results in an eigenvalue with magnitude greater than one for $J_{\text{GDA}}$, therefore all eigenvalues of $J_{\text{GDA}} - \mathbf{I}$ should be zero. We use Ky Fan inequalities which states that the sequence (in decreasing order) of the eigenvalues of $\frac{1}{2}(H + H^\top)$ majorizes the real part of the sequence of the eigenvalues of $H$ (see Moslehian (2011), page 4) for any matrix $H$. We choose $H = \frac{1}{\eta} \cdot (J_{\text{GDA}} - \mathbf{I})$. Since $R$ has both a negative and a positive eigenvalue, we get that $H$ has an eigenvalue with negative real part. The claim follows since $H$ has trace zero, thus it should have an eigenvalue with positive real part as well.

We conclude that $J_{\text{GDA}}$ has an eigenvalue with an absolute value greater than one. Using Theorem 2.2 in Daskalakis & Panageas (2018), it occurs that the set of initial conditions so that GDA converge to $(\mathbf{x}^*, \mathbf{y}^*)$ is of measure zero (for the particular choice of the step size).

## D  PROOF OF LEMMA 3.3

Firstly, we start with the min-max formulation of GMP game:

$$\min_{\mathbf{x}\in[0,1]^2} \max_{\mathbf{y}\in[0,1]^2} -x_1 x_2 y_1 y_2 - (1-x_1)(1-x_2)(1-y_1)(1-y_2) + x_1 x_2 (1-y_1)(1-y_2) +$$

$$+ (1-x_1)(1-x_2)y_1 y_2 + \omega\left(1 - x_1 x_2 - (1-x_1)(1-x_2)\right)\left(y_1 y_2 + (1-y_1)(1-y_2)\right) -$$

$$- \omega\left(1 - y_1 y_2 - (1-y_1)(1-y_2)\right)\left(x_1 x_2 + (1-x_1)(1-x_2)\right)$$

or after simplification, it is equivalent with

$$\min_{\mathbf{x}\in[0,1]^2} \max_{\mathbf{y}\in[0,1]^2} (\omega+1)(x_1+x_2)+(1-\omega)(y_1+y_2)-(x_1+x_2)(y_1+y_2)-2\omega x_1 x_2 + 2\omega y_1 y_2. \tag{D.1}$$

*Remark* D.1. Please note that the objective in the min-max is *multi-linear* and the degree of each variable in every summand is at most one (total degree is 2). Moreover note that due to the non convexity-concavity of the function above, the max-min is not equal to the min-max.

Let $(x^*_1, x^*_2, y^*_1, y^*_2)$ be a Nash equilibrium. Assuming $x^*_1, x^*_2, y^*_1, y^*_2 \in (0,1)$ from (D.1) and first-order conditions we get the system of equations

    1. $\omega + 1 - y^*_1 - y^*_2 - 2\omega x^*_2 = 0,$

2. $\omega + 1 - y_1^* - y_2^* - 2\omega x_1^* = 0,$

3. $1 - \omega - x_1^* - x_2^* + 2\omega y_2^* = 0,$

4. $1 - \omega - x_1^* - x_2^* + 2\omega y_1^* = 0.$

Combining the first two equations, we have $x_1^* = x_2^*$ and combining the last two it follows $y_1^* = y_2^*$. Dividing equation one by $\omega$ and subtracting three, it follows that $\frac{1}{\omega} - 2\frac{y_1^*}{\omega} + \omega - 2\omega y_1^* = 0$. Hence we conclude that $y_1^* = \frac{1}{2}$. As a result $y_2^* = \frac{1}{2}$ and substituting in first equation $x_1^* = x_2^* = \frac{1}{2}$.

- Assume now that $x_1^* = 0$ and $x_2^*, y_1^*, y_2^* \in (0, 1)$. Following the same idea, now only equations 2, 3, 4 hold and instead of the first we have the constraint $\omega + 1 - y_1^* - y_2^* - 2\omega x_2^* \geq 0$. From 3, 4 we conclude that $y_1^* = y_2^*$ and using 2 it holds that $y_1^* = y_2^* = \frac{1+\omega}{2}$. Using 3 follows that $1 - \omega - x_2^* + \omega(\omega + 1) = 0$. Thus $x_2^* = 1 + \omega^2 > 1$ (this is not possible because $x_2^* \in [0, 1]$).

- Consider the case that $x_1^* = 1$ and $x_2^*, y_1^*, y_2^* \in (0, 1)$. Only equations 2, 3, 4 hold and instead of the first we have the constraint $\omega + 1 - y_1^* - y_2^* - 2\omega x_2^* \leq 0$. From 3, 4 we conclude that $y_1^* = y_2^*$ and using 2 it holds that $y_1^* = y_2^* = \frac{1-\omega}{2}$. Using 3 follows that $-\omega - x_2^* + \omega(1 - \omega) = 0$. Thus $x_2^* = -\omega^2 < 0$ (this is not possible because $x_2^* \in [0, 1]$). By symmetry the same happens when $x_2^* = 0$ or $x_2^* = 1$ and $x_1^*, y_1^*, y_2^* \in (0, 1)$.

- Case $x_1^* = x_2^* = 0$ and $y_1^*, y_2^* \in (0, 1)$. Using 3, 4 we get $y_1^* = y_2^* = \frac{\omega - 1}{2\omega} < 0$ (this is not possible).

- Case $x_1^* = x_2^* = 1$ and $y_1^*, y_2^* \in (0, 1)$. Using 3, 4 we get $y_1^* = y_2^* = \frac{\omega + 1}{2\omega} > 1$ (this is not possible).

- Case $x_1^* = 0$ and $x_2^* = 1$ and $y_1^*, y_2^* \in (0, 1)$. Using 3, 4 we get $y_1^* = y_2^* = \frac{1}{2}$. Moreover one becomes $\omega - 2\omega x_2^* \geq 0$ and two $\omega - 2\omega x_1^* \leq 0$, that is $x_1^* \geq \frac{1}{2}$ and $x_2^* \leq \frac{1}{2}$ (contradiction). The case $x_1^* = 1$ and $x_2^* = 0$ and $y_1^*, y_2^* \in (0, 1)$ is symmetric.

Similarly one can consider the case where the **y** team plays pure and $x_1^*, x_2^* \in (0, 1)$. One can also check that all possible pure strategy profiles are not Nash equilibria.

## E    PROOF OF THEOREM 3.5

We split the proof in 3 parts. Before we start the proof, note that the Hessian of $U$ (E.4) has infinity norm less than 4 (since $\omega \in (0, 1)$), so $U$ has gradient Lipschitz with $L \leq 4$. Thus for the rest of the proof for GDA, we choose $\eta_{\text{OGDA}} < \frac{1}{4}$.

**GDA.** For GDA the proof will be straightforward. We will show that $(x_1^*, x_2^*, y_1^*, y_2^*)$ is a weakly Nash equilibrium. Then the claim about GDA will follow because of Lemma 3.3, Theorem, 3.2 and Remark 2.2.

Assume that player $x_1$ fixes his strategy to $x_1 = 0$. and $y_1, y_2$ keep their strategy $(\frac{1}{2}, \frac{1}{2})$. We shall show that $x_2$ is not indifferent in his support and would like to change his mixed strategy $x_2 = \frac{1}{2}$ to pure. When $x_1 = 0$ and $y_1 = y_2 = \frac{1}{2}$ the payoff of Team $A$ ($x$ variables) becomes $-\omega x_2 - 1 + \frac{\omega}{2}$. Since $\omega \in (0, 1)$, $x_2$ prefers to play $x_2 = 0$ (instead of $\frac{1}{2}$ she had). We conclude that $(\frac{1}{2}, \frac{1}{2}, \frac{1}{2}, \frac{1}{2})$ is not a weakly-stable Nash equilibrium.

**OGDA.** The Jacobian of the update rule of OGDA dynamics (use the same machinery of Section 3 in Daskalakis & Panageas (2018)) is the following:

$$J_{\text{OGDA}} = \begin{pmatrix} \mathbf{I} - 2\eta_{\text{OGDA}}\nabla^2_{\mathbf{xx}}U & -2\eta_{\text{OGDA}}\nabla^2_{\mathbf{xy}}U & \eta_{\text{OGDA}}\nabla^2_{\mathbf{xx}}U & \eta_{\text{OGDA}}\nabla^2_{\mathbf{xy}}U \\ 2\eta_{\text{OGDA}}\nabla^2_{\mathbf{yx}}U & \mathbf{I} + 2\eta_{\text{OGDA}}\nabla^2_{\mathbf{yy}}U & -\eta_{\text{OGDA}}\nabla^2_{\mathbf{yx}}U & -\eta_{\text{OGDA}}\nabla^2_{\mathbf{yy}}U \\ \mathbf{I} & \mathbf{0} & \mathbf{0} & \mathbf{0} \\ \mathbf{0} & \mathbf{I} & \mathbf{0} & \mathbf{0} \end{pmatrix}, \quad \text{(E.1)}$$

where $U$ is the payoff of Team $B$ (max) and $-U$ is the payoff of team $A$. Substituting for MPG payoff at Nash equilibrium, we have

$$\nabla^2_{\mathbf{xx}} U = \begin{pmatrix} 0 & -2\omega \\ -2\omega & 0 \end{pmatrix}, \nabla^2_{\mathbf{yy}} U = \begin{pmatrix} 0 & 2\omega \\ 2\omega & 0 \end{pmatrix}, \nabla^2_{\mathbf{xy}} U = \begin{pmatrix} -1 & -1 \\ -1 & -1 \end{pmatrix}.$$

The corresponding Jacobian matrix becomes:

$$J_{\text{OGDA}} = \begin{pmatrix} \mathbf{I} & \mathbf{0} & \mathbf{0} & \mathbf{0} \\ \mathbf{0} & \mathbf{I} & \mathbf{0} & \mathbf{0} \\ \mathbf{I} & \mathbf{0} & \mathbf{0} & \mathbf{0} \\ \mathbf{0} & \mathbf{I} & \mathbf{0} & \mathbf{0} \end{pmatrix} + \eta_{\text{OGDA}} \begin{pmatrix} 0 & 4\omega & 2 & 2 & 0 & -2\omega & -1 & -1 \\ 4\omega & 0 & 2 & 2 & -2\omega & 0 & -1 & -1 \\ -2 & -2 & 0 & 4\omega & 1 & 1 & 0 & -2\omega \\ -2 & -2 & 4\omega & 0 & 1 & 1 & -2\omega & 0 \\ 0 & 0 & 0 & 0 & 0 & 0 & 0 & 0 \\ 0 & 0 & 0 & 0 & 0 & 0 & 0 & 0 \\ 0 & 0 & 0 & 0 & 0 & 0 & 0 & 0 \\ 0 & 0 & 0 & 0 & 0 & 0 & 0 & 0 \end{pmatrix},$$

$$\tag{E.2}$$

It turns out that matrix (E.2) has characteristic polynomial (help of Mathematica)

$$\pi(\lambda) = (4(1+\omega^2)\eta^2_{\text{OGDA}}(1-2\lambda)^2 + (\lambda-1)^2\lambda^2 - 4\omega\eta_{\text{OGDA}}\lambda(1-3\lambda+2\lambda^2))$$
$$\cdot((\lambda-1)\lambda + 2\omega\eta_{\text{OGDA}}(2\lambda-1))^2.$$

$$\tag{E.3}$$

A root of $\pi(\lambda)$ is $\frac{1}{2}\left(1 + 4\eta_{\text{OGDA}}j + 4\omega\eta_{\text{OGDA}} + \sqrt{1 - 16\eta^2_{\text{OGDA}} + 32j\omega\eta^2_{\text{OGDA}} + 16\omega^2\eta^2_{\text{OGDA}}}\right)$, which is in absolute value greater than one for $0 < \eta_{\text{OGDA}} \le \omega$ It is also easy to see the Hessian of $U$, that is:

$$\begin{pmatrix} \nabla^2_{\mathbf{xx}} U & \nabla^2_{\mathbf{xy}} U \\ \nabla^2_{\mathbf{yx}} U & \nabla^2_{\mathbf{yy}} U \end{pmatrix} = \begin{pmatrix} 0 & -2\omega & -1 & -1 \\ -2\omega & 0 & -1 & -1 \\ -1 & -1 & 0 & 2\omega \\ -1 & -1 & 2\omega & 0 \end{pmatrix}$$

$$\tag{E.4}$$

is invertible for $\omega \in (0,1)$ (this is true as the eigenvalues are $-2\omega, 2\omega, -2\sqrt{1+\omega^2}, 2\sqrt{1+\omega^2}$, non of which is zero). From Theorem 3.2 in Daskalakis & Panageas (2018) follows that the initial conditions so that OGDA converges to the Nash equilibrium is of measure zero for step size $\eta_{\text{OGDA}} < \frac{1}{2L} \le \frac{1}{8}$ (where $L$ is the Lipschitz constant of $\nabla U$). We choose $\eta_{\text{OGDA}} \le \min(\frac{1}{8}, \omega)$ and the claim follows.

**EG.** The Jacobian of EG dynamics computed at the Nash equilibrium (fixed point) is given below (made use of chain rule):

$$J_{\text{EG}} = \mathbf{I} + \eta_{\text{EG}} \begin{pmatrix} -\nabla^2_{\mathbf{xx}} U(\mathbf{x}^*, \mathbf{y}^*) & -\nabla^2_{\mathbf{xy}} U(\mathbf{x}^*, \mathbf{y}^*) \\ \nabla^2_{\mathbf{yx}} U(\mathbf{x}^*, \mathbf{y}^*) & \nabla^2_{\mathbf{yy}} U(\mathbf{x}^*, \mathbf{y}^*) \end{pmatrix} + \eta^2_{\text{EG}} \begin{pmatrix} -\nabla^2_{\mathbf{xx}} U(\mathbf{x}^*, \mathbf{y}^*) & -\nabla^2_{\mathbf{xy}} U(\mathbf{x}^*, \mathbf{y}^*) \\ \nabla^2_{\mathbf{yx}} U(\mathbf{x}^*, \mathbf{y}^*) & \nabla^2_{\mathbf{yy}} U(\mathbf{x}^*, \mathbf{y}^*) \end{pmatrix}^2.$$

$$\tag{E.5}$$

We substitute with the values and we get

$$J_{\text{EG}} = \mathbf{I} + \eta_{\text{EG}} \begin{pmatrix} 0 & 2\omega & 1 & 1 \\ 2\omega & 0 & 1 & 1 \\ -1 & -1 & 0 & 2\omega \\ -1 & -1 & 2\omega & 0 \end{pmatrix} + \eta^2_{\text{EG}} \begin{pmatrix} 4\omega^2 - 2 & -2 & 4\omega & 4\omega \\ -2 & 4\omega^2 - 2 & 4\omega & 4\omega \\ -4\omega & -4\omega & -2+4\omega^2 & -2 \\ -4\omega & -4\omega & -2 & -2+4\omega^2 \end{pmatrix}.$$

$$\tag{E.6}$$

The eigenvalues of $J_{\text{EG}}$ are $1 + \eta_{\text{EG}}(-2\omega + 4\eta_{\text{EG}}\omega^2)$, $1 + \eta_{\text{EG}}(-2\omega + 4\eta_{\text{EG}}\omega^2)$, $1 + \eta_{\text{EG}}\left(-4\eta_{\text{EG}} + 2\omega + 4\eta_{\text{EG}}\omega^2 - 2\sqrt{-1 - 8\eta_{\text{EG}}\omega - 16\eta^2_{\text{EG}}\omega^2}\right)$, and $1 + \eta_{\text{EG}}\left(-4\eta_{\text{EG}} + 2\omega + 4\eta_{\text{EG}}\omega^2 + 2\sqrt{-1 - 8\eta_{\text{EG}}\omega - 16\eta^2_{\text{EG}}\omega^2}\right)$. If $-4\eta_{\text{EG}} + 2\omega + 4\eta_{\text{EG}}\omega^2 > 0$ then $J_{\text{EG}}$ will have an eigenvalue with absolute value greater than one. A sufficient condition for that is $\eta_{\text{EG}} \le \frac{\omega}{2}$. Finally, for the same choice of stepsizes we have that $J_{\text{EG}}$ is invertible for all $(\mathbf{x}, \mathbf{y})$, hence the $EG$ dynamics is a local diffeomorphism. From Theorem 2 in arxiv version of Lee et al. (2019) the claim for EG method follows.

**OMWU.** The Jacobian of the update rule of OMWU dynamics (use the same machinery of Section 3 in Daskalakis & Panageas (2019)) is the following:

$$J_{\text{OMWU}} = \begin{pmatrix} 1 & \eta\omega & \eta & \eta & 0 & -\frac{\eta\omega}{2} & -\frac{\eta}{2} & -\frac{\eta}{2} \\ \eta\omega & 1 & \eta & \eta & -\frac{\eta\omega}{2} & 0 & -\frac{\eta}{2} & -\frac{\eta}{2} \\ -\eta & -\eta & 1 & \eta\omega & \frac{\eta}{2} & \frac{\eta}{2} & 0 & -\frac{\eta\omega}{2} \\ -\eta & -\eta & \eta\omega & 1 & \frac{\eta}{2} & \frac{\eta}{2} & -\frac{\eta\omega}{2} & 0 \\ 1 & 0 & 0 & 0 & 0 & 0 & 0 & 0 \\ 0 & 1 & 0 & 0 & 0 & 0 & 0 & 0 \\ 0 & 0 & 1 & 0 & 0 & 0 & 0 & 0 \\ 0 & 0 & 0 & 1 & 0 & 0 & 0 & 0 \end{pmatrix}. \tag{E.7}$$

It turns out that matrix (E.7) has characteristic polynomial (help of Mathematica)

$$\pi(\lambda) = \frac{1}{16} \left( (4 + \omega^2)\eta^2(1 - 2\lambda)^2 + 4(\lambda - 1)^2\lambda^2 - 4\omega\eta\lambda(1 - 3\lambda + 2\lambda^2) \right) \cdot \tag{E.8}$$
$$(2(\lambda - 1)\lambda + \omega\eta(2\lambda - 1))^2 .$$

One root of the polynomial $\pi(\lambda)$ above is $\frac{1}{2}(1 + 2\eta j + \eta\omega + \sqrt{1 - 4\eta^2 + 4\eta^2\omega j + \eta^2\omega^2})$. As in the analysis of OGDA, it turns out that for $0 < \eta < \omega$ the aforementioned root has absolute value greater than one.

Therefore we conclude that the Nash equilibrium where each agent plays $(\frac{1}{2}, \frac{1}{2})$ is repelling (and thus the fact that the set of initial conditions so that OMWU converges to that particular fixed point is of measure zero is derived from standard arguments from Lee et al. (2019). To conclude the proof we need to exclude that OMWU will stabilize in other fixed points. Note that OMWU can stabilize to points that are not Nash equilibria (e.g., all pure strategy profiles are fixed points). To exclude such stabilization, the trick is that if OMWU stabilizes to a point, it should be (approximate coarse correlated equilibrium where the approximation depends on the size of the stepsize). However, none of the pure strategy profiles is a $\omega/2$- approximate coarse correlated equilibrium, so OMWU does not stabilize. Hence if we choose $0 < \eta < \omega/2$, OMWU does not stabilize.

## F    PROOF OF THEOREM 3.8

We first compute the Jacobian of KPV-GDA dynamics (2.2) where we have eliminated one variable per agent (to avoid using the projection operator). The Jacobian has the following form:

$$J_{\text{KPV-GDA}} = \mathbf{I} + \eta \begin{pmatrix} k \cdot \mathbf{I} + \begin{pmatrix} -\nabla^2_{\mathbf{xx}} U(\mathbf{x}^*, \mathbf{y}^*) & -\nabla^2_{\mathbf{yx}} U(\mathbf{x}^*, \mathbf{y}^*) \\ \nabla^2_{\mathbf{xy}} U(\mathbf{x}^*, \mathbf{y}^*) & \nabla^2_{\mathbf{yy}} U(\mathbf{x}^*, \mathbf{y}^*) \end{pmatrix} & -k \cdot \mathbf{I} \\ p \cdot \mathbf{I} & -p \cdot \mathbf{I} \end{pmatrix}. \tag{F.1}$$

It suffices to show that the spectral radius of $J_{\text{KPV-GDA}}$ is less than one for step size $\eta$ small enough. We first show the claim below

**Claim F.1.** *Let $J = I + \eta M$ where $I$ is the identity matrix. Suppose that $M$ has all of its eigenvalues with a real part that is negative. Then, there exists an interval $(0, \eta_0)$ so that if $\eta \in (0, \eta_0)$ then $J$ has all of its eigenvalues with an absolute value less than one.*

*Proof.* Let $a_i + b_i \cdot j$ be an eigenvalue of $M$ such that $a_i < 0$. Assume that $\eta < -\frac{2a_i}{a_i^2 + b_i^2}$, then the corresponding eigenvalue of $J$ is $1 + \eta(a_i + b_i \cdot j)$, the magnitude of which is $(1 + \eta a_i)^2 + \eta^2 b_i^2 = 1 + 2\eta a_i + \eta^2(a_i^2 + b_i^2) < 1$ since $-2a_i > \eta(a_i^2 + b_i^2)$ (by assumption). Hence we can choose $\eta_0$ to be $\min_i \frac{-2a_i}{a_i^2 + b_i^2} > 0$. $\qquad\square$

Using Claim F.1, we conclude that as long as the matrix $M$ below has eigenvalues with real part negative then $(x^*, y^*)$ is attracting:

$$M := \begin{pmatrix} k \cdot \mathbf{I} + \begin{pmatrix} -\nabla^2_{\mathbf{xx}} U(\mathbf{x}^*, \mathbf{y}^*) & -\nabla^2_{\mathbf{yx}} U(\mathbf{x}^*, \mathbf{y}^*) \\ \nabla^2_{\mathbf{xy}} U(\mathbf{x}^*, \mathbf{y}^*) & \nabla^2_{\mathbf{yy}} U(\mathbf{x}^*, \mathbf{y}^*) \end{pmatrix} & -k \cdot \mathbf{I} \\ p \cdot \mathbf{I} & -p \cdot \mathbf{I} \end{pmatrix}. \tag{F.2}$$

By setting $M_{11}, M_{12}, M_{21}, M_{22}$ as the corresponding block matrices we compute the characteristic polynomial of $M$. It holds

$$\det(M - \lambda \mathbf{I}) = \det\left(\left[\begin{array}{c|c} M_{11} & M_{12} \\ \hline M_{21} & M_{22} \end{array}\right] - \lambda \mathbf{I}\right) = \det\left(\left[\begin{array}{c|c} M_{11} - \lambda \mathbf{I} & -k\mathbf{I} \\ \hline p\mathbf{I} & -p\mathbf{I} - \lambda \mathbf{I} \end{array}\right]\right) = \quad \text{(F.3)}$$

$$= \det\left(\left[\begin{array}{c|c} M_{11} - \lambda \mathbf{I} - \frac{kp}{p+\lambda}\mathbf{I} & -k\mathbf{I} \\ \hline 0 & -p\mathbf{I} - \lambda \mathbf{I} \end{array}\right]\right) = \quad \text{(F.4)}$$

$$= \det\left(M_{11} - \left(\lambda + \frac{kp}{p+\lambda}\right)\mathbf{I}\right)\det\left(-(p+\lambda)\mathbf{I}\right). \quad \text{(F.5)}$$

$\det(M - \lambda\mathbf{I})$ has roots at $\lambda = -p$ and when

$$\det\left(M_{11} - \left(\lambda + \frac{kp}{p+\lambda}\right)\mathbf{I}\right) = 0.$$

Therefore, if $M$ has all eigenvalues with real part negative, we must have $p > 0$. Let $\rho$ be an eigenvalue of $\begin{pmatrix} -\nabla^2_{\mathbf{xx}}U(\mathbf{x}^*, \mathbf{y}^*) & -\nabla^2_{\mathbf{yx}}U(\mathbf{x}^*, \mathbf{y}^*) \\ \nabla^2_{\mathbf{xy}}U(\mathbf{x}^*, \mathbf{y}^*) & \nabla^2_{\mathbf{yy}}U(\mathbf{x}^*, \mathbf{y}^*) \end{pmatrix}$. It must hold that

$$\lambda + \frac{kp}{p+\lambda} - k = \rho. \quad \text{(F.6)}$$

We need to investigate under what assumptions $\lambda$ will have real part negative. We expand (F.6) and we get

$$\lambda^2 + \lambda(p - k - \rho) - \rho p = 0. \quad \text{(F.7)}$$

We solve Equation (F.7) and we get

$$\lambda_{1,2} = \frac{-p + k + \rho \pm \sqrt{(p - k - \rho)^2 + 4\rho p}}{2}. \quad \text{(F.8)}$$

We provide sufficient conditions so that $\text{Re}(\lambda_{1,2}) < 0$. We consider the following cases:

- $\rho$ is real and negative. In this case observe that for $p = 0$ we have that $\lambda_{1,2} = k + \rho$ and $0$. Note that for $k < 0$, the first eigenvalue has real part negative and the second eigenvalue is zero. We compute the first derivative at $p = 0$ and this gives $\frac{1}{2}\left(-1 + \frac{-k+\rho}{|k+\rho|}\right) = \frac{\rho}{-k-\rho} < 0$. Therefore the $\lambda_{1,2}$ as a function of $p$ is strictly decreasing at zero. We conclude that for $p$ sufficiently small and positive, both eigenvalues of $M$ will be real and negative. Hence if an eigenvalue of $H$ is real and negative then for $p$ sufficiently small positive and $k < 0$ both eigenvalues of $M$ will be negative.

- $\rho$ is complex with $|\text{Im}(\rho)| \neq 0$. In this case observe that for $p = 0$ we have that $\lambda_{1,2} = k + \rho$ and $0$. If we choose $\min(0, -\text{Re}(\rho)) > k$, then the first eigenvalue will have real part negative. Now using the same idea as for the first case, we compute the first derivative of the real part of $\lambda_{1,2}$ as a function of $p$ and we get

$$\frac{1}{2}\left(-1 \pm \frac{k^2 - \text{Re}(\rho)^2 - \text{Im}(\rho)^2}{(k + \text{Re}(\rho))^2 + \text{Im}(\rho)^2}\right).$$

The expression above is negative as long as $-\text{Re}(\rho)^2 - \text{Im}(\rho)^2 - k\text{Re}(\rho) < 0$. The equation is trivially satisfied when $\text{Re}(\rho) < 0$ since $k$ is chosen to be negative. Assume $\text{Re}(\rho) > 0$ (if it is zero then the above is trivially true since the imaginary part is non-zero). It occurs that the inequality above is true when $k > -\frac{|\rho|^2}{\text{Re}(\rho)}$. Since $k$ is chosen to be smaller than $\min(0, -\text{Re}(\rho))$, it suffices to show that

$$-\text{Re}(\rho)) > -\frac{|\rho|^2}{\text{Re}(\rho)} \text{ is satisfied by our assumptions.}$$

The above is true as long as $\text{Im}(\rho) \neq 0$. Let $E$ be the set of eigenvalues $\rho$ of $H$ with real part positive (and non-zero imaginary part by assumption). There exists a choice for $\eta, k, p$, in which $\eta, p > 0$ and sufficiently small and $k$ is negative and chosen to be

$$\min_{\rho \in E} \frac{|\rho|^2}{\text{Re}(\rho)} > -k > \max_{\rho \in E} \text{Re}(\rho).$$

# G    CIFAR-10

CIFAR-10 is a well-established testbed for various GAN architectures. It contains 10 balanced classes of images of different objects.

Just to further illustrate the merits of multi-agent GANs, we offer a selection of images generated by WGAN-GP Arjovsky et al. (2017) and the MGAN Hoang et al. (2017).

In the pictures provided by the WGAN-GP we see that across iterations, the generated samples tend to cover only certain classes of the dataset while it takes longer for the samples to become realistic.

On the other hand, the MGAN architecture from an early stage provides with diverse samples that tend to be more realistic from early on.

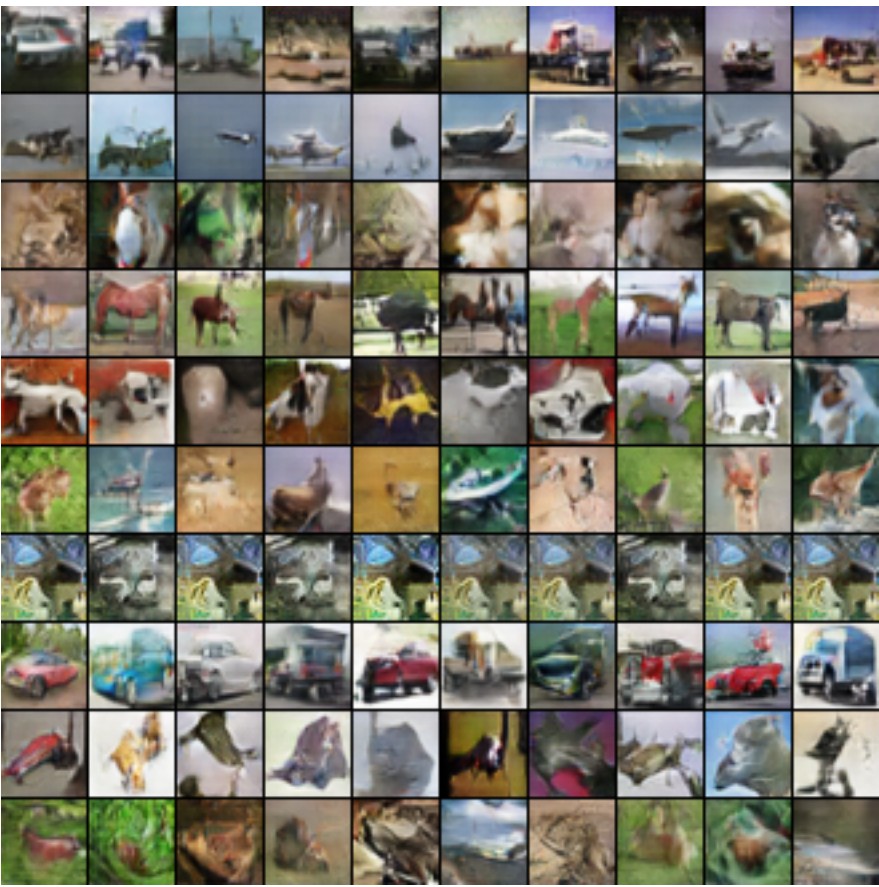

Figure 6: MGAN after 300 iterations

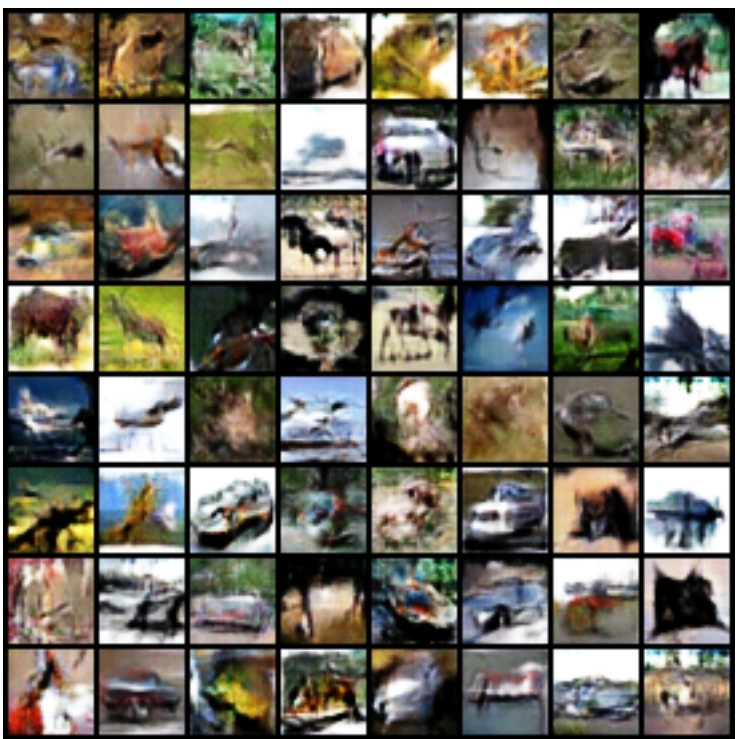

Figure 7: WGAN after 10000 iterations

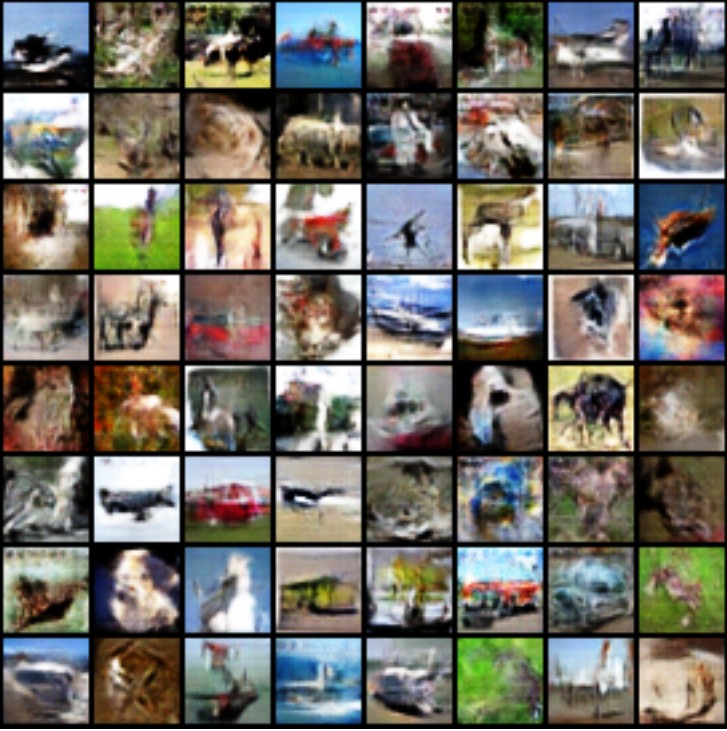

Figure 8: WGAN after 15000 iterations

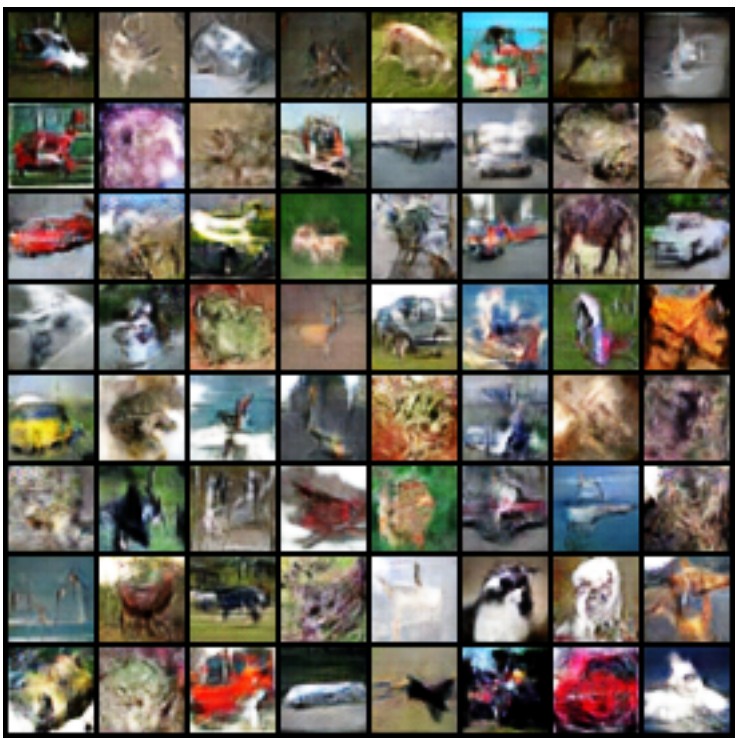

Figure 9: WGAN after 25000 iterations

