# OpenReview forum: "Teamwork makes von Neumann work:Min-Max Optimization in Two-Team Zero-Sum Games"
_ICLR.cc/2022/Conference — ICLR 2022 Submitted_

### Official Review · Reviewer_FpwP · 2021-10-25

**Correctness:** 4
**Technical Novelty And Significance:** 1
**Empirical Novelty And Significance:** 1
**Recommendation:** 3
**Confidence:** 4

**Main Review:**

The claims in this paper are supported by theoretical and experimental results.

However, my concerns are:

The solution concept for the studied two-team zero-sum game is Nash equilibrium, and this two-team zero-sum game is an N-player game. It is well-known that computing a Nash equilibrium in N-player games is PPAD-complete (Daskalakis et al., 2009) (i.e.., it is unlikely to have a polynomial-time algorithm for computing a Nash equilibrium), and this paper shows that finding a Nash equilibrium in two-team zero-sum games is CLS-hard (i.e., it is unlikely to have a polynomial-time algorithm for computing a Nash equilibrium).

Gradient-based algorithms have been used to compute a Nash equilibrium in N-player games (Mckelvey, 1988; Mckelvey et al., 2014; Gemp et al., 2021), which cannot guarantee the convergence in general but will converge to a Nash equilibrium with special settings. That is the similar result presented in this paper.

Therefore, this paper cannot make us understand two-team zero-sum games better.



McKelvey, R.D., 1998. A Liapunov function for Nash equilibria.\
McKelvey, Richard D., McLennan, Andrew M., and Turocy, Theodore L. (2014). Gambit: Software Tools for Game Theory, Version 16.0.1. http://www.gambit-project.org. \
Gemp, I., Savani, R., Lanctot, M., Bachrach, Y., Anthony, T., Everett, R., Tacchetti, A., Eccles, T. and Kramár, J., 2021. Sample-based Approximation of Nash in Large Many-Player Games via Gradient Descent. arXiv preprint arXiv:2106.01285.


The presentation can be improved:
1.	It will be better to explain the given equations, e.g., the gradient formulation in Section 2. By the way, what is k+1/2 in Extra Gradient Method?
2.	How to obtain Eq.(2.5) from Eq.(2.4)? What is \mu in Eq.(2.5)?
3.	What is “measure zero” in Theorem 3.2?
4.	The number in figures is unclear.
5.	Need to explain how the experimental results in Section 4 validate the theoretical findings (or which finding).


Minor: \
There are many grammar issues:\
In the abstract: “Moreover In ”\
In Section 1: “Firstly, computing local Nash Equilibria (NE) in general non-convex non-concave games **are** PPAD-complete”, "MD-GAN (Hardy et al., 2019) **are**"\
In Section 3.2: “we prove an important **Theorem** ”\
In Section 3.3: “**course**-correlated equilibria (CCE)”\
In Section 4: “On the other hand our architecture with a small number of neurons achieves to **We
defer an execution** of multi-generators multi-discriminators architectures for CIFAR-10 **again to** the
paper’s supplement. ”

About the related work on TME, the following two latest papers should be discussed:\
Zhang, Y., An, B. and Černý, J., 2021, May. Computing Ex Ante Coordinated Team-Maxmin Equilibria in Zero-Sum Multiplayer Extensive-Form Games. In Proceedings of the AAAI Conference on Artificial Intelligence (Vol. 35, No. 6, pp. 5813-5821).\
Farina, G., Celli, A., Gatti, N. and Sandholm, T., 2021, July. Connecting Optimal Ex-Ante Collusion in Teams to Extensive-Form Correlation: Faster Algorithms and Positive Complexity Results. In International Conference on Machine Learning (pp. 3164-3173). PMLR.


**Summary Of The Paper:**

This paper shows that finding a Nash equilibrium in team zero-sum games is CLS-hard (i.e., it is unlikely to have a polynomial-time algorithm for computing a Nash equilibrium).  They show that the commonly used gradient-based methods cannot converge to a Nash equilibrium and then propose a gradient-based method with special conditions, which manages to stabilize around Nash equilibria.

**Summary Of The Review:**

This paper cannot make us understand two-team zero-sum games better.

***After the discussion:***
It seems that the authors have misunderstood the relation between different solution concepts.

The statement “the notion of"per player Nash equilibria" can be seen as a smoother figure of merit for the performance of a defensive team against an adversary” is not correct because a per-player Nash equilibrium may not be a team maxmin equilibrium (TME). Similar to TME, a two-team maximin equilibrium is still a per-player Nash equilibrium, but a per-player Nash equilibrium may not be a two-team maximin equilibrium. This paper focuses on min-max optimization in team zero-sum games, but it only achieves a per-player Nash equilibrium. I think this is inappropriate.

---

> ### Author Response · Authors · 2021-11-23
> **Response to Reviewer FpwP (0/5)**
>
> Thank you, our detailed response follows:

---

> > ### Author Response · Authors · 2021-11-23
> > **Response to Reviewer FpwP (5/5)**
> >
> > ## Measure-zero:
> > The Lebesgue measure gives a concrete way to measure the volume (or area) of subsets of $\mathbb{R}^n$. For simplicity, we will only discuss the special case about sets that have Lebesgue-measure zero. Intuitively, these sets are "so small that can not be observed".
> >
> > Let $n$ be a natural number and $\mathbb{R}^n$ be the standard Euclidean space.
> > Then an
> > open cube in $\mathbb{R}^n$ is a product of open intervals $U=(a_1,b_1)\times(a_2,b_2)\times\cdots\times (a_n,b_n)$ and the volume of this cube is defined to be $vol (U) = (b_1−a_1)\times(b_2−a_2)\cdots(b_n−a_n)$.
> >
> >
> > Now, in mathematical analysis, a null set  $\displaystyle A\subset \mathbb {R} $ is a measurable set that has measure zero. This can be characterized as a set that can be covered by a countable union of intervals of arbitrarily small total length. Formally speaking the set $A$ is measure-zero
> > $ \forall \varepsilon > 0, \ \exists \left\\{U_n\right\\}_n :$
> >
> > $$ U_n=(a_n,b_n)\subset \mathbb{R}: \quad
> > A \subset \bigcup_{n = 1}^{\infty} U_n \ \land\ \sum_{n = 1}^\infty \left|U_n\right| < \varepsilon \,
> > $$
> >
> > The notion of the null set should not be confused with the empty set as defined in set theory. Although the empty set has Lebesgue measure zero, there are also non-empty sets that are null. For example, any non-empty countable set of real numbers has Lebesgue measure zero and therefore is null.
> >
> > In the optimization literature, we use the phrase of *the set of initializations for which GDA converges to $(x^\star,y^\star)$ is of measure zero* to explain that almost always GDA will diverge except some pathological cases.
> >
> > From a probability theory perspective, this also holds true to any probability measure absolutely continuous w.r.t.   the Lebesgue measure (e.g. any continuous probability distribution).   That is,$\Pr(\lim_{t} x^{GDA}_t=x^\star) = 0 $ where randomness is about the initialization of the method.
> >
> > ## Figures
> > We apologize for the quality of the images. A wrong choice of parameters in latex code, unfortunately, diminished their resolution. We update them with a better resolution.
> >
> > ## Relationship of theoretical results to presented experiments
> > The theory of competitive learning is yet nascent when compared to their "single-agent" counterparts. Nevertheless, we feel that it is crucial to highlight the advantages that this regime of learning holds through applications already present in the literature.
> > We aspire to contribute to the theory of existing multi-agent applications in machine learning and we believe to have done so by both improving the understanding of the utility landscape (w.r.t. to every agent) as well as the dynamics induced by common optimization algorithms upon it. We also intend to use our proposed algorithm in multi-agent gan optimization, an experiment that in itself should require a different article.
> >
> > Thank you again for your time, we hope that our clarifications and revision are satisfactory.
> >
> > With kind regards,\
> > *The authors*
> >
> > [1] Panayotis Mertikopoulos, Christos H. Papadimitriou, and Georgios Piliouras.
> > Cycles in adversarial regularized learning. In Proceedings of the Twenty-Ninth
> > Annual ACM-SIAM Symposium on Discrete Algorithms, SODA 2018, New Orleans, LA,
> > USA, January 7-10, 2018, pages 2703–2717, 2018
> >
> > [2] Constantinos Daskalakis, Andrew Ilyas, Vasilis Syrgkanis, and Haoyang Zeng.
> > Training GANs with optimism. In ICLR, 2018
> >
> > [3] Jacob Abernethy, Kevin A Lai, and Andre Wibisono. Last-iterate convergence rates
> > for min-max optimization. arXiv preprint arXiv:1906.02027, 2019.
> >
> > [4] Panayotis Mertikopoulos, Bruno Lecouat, Houssam Zenati, Chuan-Sheng Foo,
> > Vijay Chandrasekhar, and Georgios Piliouras. Optimistic mirror descent in
> > saddle-point problems: Going the extra(-gradient) mile. In ICLR, 2019
> >
> > [5] Aryan Mokhtari, Asuman Ozdaglar, and Sarath Pattathil. A unified analysis of
> > extra-gradient and optimistic gradient methods for saddle point problems:
> > Proximal point approach. In International Conference on Artificial Intelligence and
> > Statistics, pages 1497–1507. PMLR, 2020

---

> > > ### Comment · Reviewer_FpwP · 2021-11-25
> > > **Thanks for your response**
> > >
> > > Thanks for your response.
> > >
> > > My main concerns is that: this paper only computes a multiplayer Nash equilibrium (see the solution concept definition after Eq.(2.1)) in a two-team zero-sum game, which may not be a max-min equilibrium for two teams (one team's per-player strategy maximizes the team's utility that the other team's per-player strategy minimizes). It seems that this paper just computes a multiplayer Nash equilibrium in a special game, and its results do not surprise me.

---

> > > > ### Author Response · Authors · 2021-11-27
> > > > **Further response (2/2)**
> > > >
> > > > ### The element of surprise
> > > >
> > > > * The fact that algorithms like the Extragradient Method, Optimistic Gradient Descent-Ascent **fail** in this family of games would be surprising to most folks in the community of $\min\max$ optimization since their merits have been championed and found tremendous success and widespread applications from two-player zero-sum games to reinforcement learning, granting even last-iterate convergence in those tasks. Generally, discretizations of the Gradient Flow and Replicator Dynamics are our main way of tackling such problems.
> > > >
> > > >
> > > > * On top of the failure proof, we also propose an **algorithm** that **does converge in the last iterate**. Our hardness result hints also that the order of magnitude of the number of iterations it should take for it to converge is $\Omega \big( \mathrm{poly} \log (\frac{1}{\epsilon} ) \big)$.
> > > >
> > > >
> > > > * Our proposed algorithm is able to compute a Nash Equilibrium in **any two-team game** with a very simple parametrization when a sufficient condition is met. The claim that we do so in just a "special game" is an utter misunderstanding. The same idea could also be used in nonconvex optimization settings.
> > > >
> > > > Feeling surprised or not, by being utterly subjective, does not suggest a criterion for assessing a scientific article at least not by the standards of ICLR as far as we know.
> > > >
> > > > At the end of the day, PPAD-completeness of approximate Nash was not surprising since the class was defined by a conceptual imitation of Lemke Howson Dynamics. However, it was groundbreaking, similarly with Popov's Optimistic or Koprelevich's Extragradient. Hence, expecting an element of surprise or measuring an emotional response is not a proper way to assess scientific work.
> > > >
> > > > We honestly hope that our attempt for answering in detail your concerns will shift your opinion.
> > > >
> > > > [1] Tim Roughgarden. Intrinsic robustness of the price of anarchy. In Proceedings of the forty-first
> > > > annual ACM symposium on Theory of computing, pp. 513–522, 2009
> > > >
> > > > [2] Youzhi Zhang, Bo An, and Jakub Cerny. Computing ex ante coordinated team-maxmin equilibria
> > > > in zero-sum multiplayer extensive-form games. arXiv preprint arXiv:2009.12629, 2020.
> > > >
> > > > [3] Joel Z Leibo, Vinicius Zambaldi, Marc Lanctot, Janusz Marecki, and Thore Graepel. Multi-agent
> > > > reinforcement learning in sequential social dilemmas. arXiv preprint arXiv:1702.03037, 2017
> > > >
> > > > [4] Yongcan Cao, Wenwu Yu, Wei Ren, and Guanrong Chen. An overview of recent progress in the
> > > > study of distributed multi-agent coordination. IEEE Transactions on Industrial informatics, 9(1):
> > > > 427–438, 2012.
> > > >
> > > > [5] Hassam Ullah Sheikh, Mina Razghandi, and Ladislau Boloni. Learning distributed cooperative
> > > > policies for security games via deep reinforcement learning. In 2019 IEEE 43rd Annual Computer
> > > > Software and Applications Conference (COMPSAC), volume 1, pp. 489–494. IEEE, 2019.

---

> > > > > ### Comment · Reviewer_FpwP · 2021-11-28
> > > > > **Thanks for your response**
> > > > >
> > > > > Thanks for your response.
> > > > >
> > > > > I think the authors have misunderstood my statements. I did not suggest that you should use TMECor as the solution concept. A two-team maximin equilibrium is still a per-player Nash equilibrium.
> > > > >
> > > > > It seems that the authors have misunderstood the relation between different solution concepts.
> > > > >
> > > > > The statement “the notion of"per player Nash equilibria" can be seen as a smoother figure of merit for the performance of a defensive team against an adversary” is not correct because a per-player Nash equilibrium may not be a team maxmin equilibrium (TME). Similar to TME, a two-team maximin equilibrium is still a per-player Nash equilibrium, but a per-player Nash equilibrium may not be a two-team maximin equilibrium. This paper focuses on min-max optimization in team zero-sum games, but it only achieves a per-player Nash equilibrium. I think this is inappropriate.
> > > > >
> > > > > "no surprise" is related to the related work I mentioned.

---

> > > > > > ### Author Response · Authors · 2021-11-29
> > > > > > **Further clarifications**
> > > > > >
> > > > > > As stated in [1]:
> > > > > >
> > > > > > `A Team–maxmin equilibrium is a Nash equilibrium with the properties of being unique (except for degeneracies)
> > > > > > and the best one for the team.`
> > > > > >
> > > > > > **Our negative results** (GDA, OGDA, EG, and OMWU fail to converge) hold for any (mixed) Nash Equilibrium, hence they **carry over to Team-maxmin equilibria** since the latter remain NE. This is trivial since Nash Equilibria are a super-set of the latter.
> > > > > >
> > > > > > Furthermore, computing a Team Maxmin equilibrium (not any NE), again in [1]:
> > > > > > `When the number of players is given, finding a Team–maxmin equilibrium is` **FNP–hard** `and the Team–maxmin value is inapproximable in additive sense.`
> > > > > > So asking us to compute such equilibria with local search algorithms is quite unfair a demand, to say the least.
> > > > > >
> > > > > > Both papers you provided explicitly showcase reasons as to why they chose to dismiss TME (with FNP-hardness being one of them) as a solution concept in favor of TMECor.
> > > > > >
> > > > > > But, we do **provide an algorithm** that converges to Nash Equilibria in two-team games which is the **next best thing** if coordination is out of the question. We remind you that your assessment that our work does so only in a special game was rather hasty but you chose not to revoke your claim.
> > > > > >
> > > > > > We sadly have to admit that the conversation circulates around concerns that are not stated clearly. Your initial concern was complexity-related while you ask that we discuss **TME** and provide 2 references which both discuss **Ex-Ante Correlated Team Maxmin** Equilibria. After we addressed your concerns on the matter of complexity and the matter of **TMECor** and why we did not choose it, you pointed out that we should be focusing on a **different kind** of equilibrium, namely **TME**. Even as such, our results trivially carry over to TMEs.
> > > > > >
> > > > > >
> > > > > > [1] Basilico, N., Celli, A., De Nittis, G. and Gatti, N., 2017, February. Team-maxmin equilibrium: efficiency bounds and algorithms. In Proceedings of the AAAI Conference on Artificial Intelligence (Vol. 31, No. 1).
> > > > > >
> > > > > > [2] Zhang, Y., An, B. and Černý, J., 2021, May. Computing Ex Ante Coordinated Team-Maxmin Equilibria in Zero-Sum Multiplayer Extensive-Form Games. In Proceedings of the AAAI Conference on Artificial Intelligence (Vol. 35, No. 6, pp. 5813-5821).
> > > > > >
> > > > > > [3] Farina, G., Celli, A., Gatti, N. and Sandholm, T., 2021, July. Connecting Optimal Ex-Ante Collusion in Teams to Extensive-Form Correlation: Faster Algorithms and Positive Complexity Results. In International Conference on Machine Learning (pp. 3164-3173). PMLR.

---

> > > > ### Author Response · Authors · 2021-11-27
> > > > **Further response (1/2)**
> > > >
> > > > ### Flavors of equilibria
> > > >
> > > > In the revised version of our paper, we did include the reasons as to why we chose the equilibrium concept of the *per-player* Nash Equilibrium rather than the ones you suggest, but we have no problem demonstrating them once more. There is a certain rationale that no matter what someone's personal taste is in flavors of equilibria that stems from a pragmatistic view on AI problems.
> > > >
> > > > The reasons are specific and purposeful.
> > > >
> > > > Initially, it is easy to see that TMECor similarly with the Coarse correlated equilibria can achieve better social welfare but as in the case of congestion games, found in literature, their price of anarchy can become significantly worse [1]. In order to give a better result, TMECor requests an a priori knowledge of the game. For example, as the paper you suggested [2] mentioned:
> > > >
> > > > `For example, in multiplayer poker games, a team may play against an adversary player, but they cannot communicate and discuss their strategy during the game due to the rule.`
> > > >
> > > > **But...**
> > > > * In **contemporary challenges for AI agents** (but, also financial agents that form alliances), **prior knowledge of the game** does **not always** hold. Furthermore, there are settings in which the **players do not know in advance** who **their teammates** are. We want to address such settings and the equilibrium concept that you propose:
> > > >     1. *fails* to address not knowing the game a-priori even if there exists exact knowledge of the team
> > > >     2. *fails* to address not knowing who your teammates are even if prior knowledge of the game holds
> > > >
> > > > **So...**
> > > >
> > > > * In **our model**:
> > > >     1. Every **player** decides a priori only with respect to **his/her own dynamics**
> > > >     2. There **no need** to **communicate** with their teammates.
> > > >
> > > > Problems of this kind are prevalent in modern ML applications like strategic conflict resolution [3], coordination between autonomous vehicles [4] , or collaboration of agents in defensive escort teams [5]. In this kind of games, each agent is simultaneously working towards maximizing its own payoff (local reward) as well as the collective success of the team (global reward).
> > > >
> > > >
> > > > The notion of per player NE expresses the model hurdles both simultaneous team intra-collaboration and inter-competition and does **not** **assume/require** coordination and/or **communication**.
> > > >
> > > >
> > > > Finally, one more reason, the notion of "per player Nash equilibria" can be seen as a smoother figure of merit for the performance of a defensive team against an adversary. Notably, in the vast majority of the aforementioned literature, a single super-powerful adversary has been used (hence, they do not concern *two*-team games as our work does).

---

> > ### Author Response · Authors · 2021-11-23
> > **Response to Reviewer FpwP (4/5)**
> >
> > So we can apply the following manipulations in the objective (The notation $[x_i^2]$ stands for the vector that has as entries the values of the vector $\mathbf{x}$ squared):
> >
> > $$
> >      \max_{ \mathbf{v}, \mathbf{w}} \min_{\boldsymbol{\theta}, p}
> >            \mathbb{E}_\text{$\mathbf{y} \sim real$} \Big[ D_\mathbf{v} (\mathbf{y} ) +D_\mathbf{w}( \mathbf{y} ) \Big]
> >            -\\
> >             - \mathbb{E}_\text{$ \mathbf{z} \sim \mathcal{N}(0,\mathbf{I} ), \zeta \sim \mathcal{N}(0,1) $}
> >             \Big[ G_p(\zeta)\cdot
> >                 \Big( D_\mathbf{v}\big( - G_\boldsymbol{\theta}( \mathbf{y})\big)+D_\mathbf{w}\big(G_\boldsymbol{\theta}( \mathbf{y})\big) \Big)
> >             \\
> >               + \big(1-G_p(\zeta)\big)\cdot \Big(D_\mathbf{v}\big(G_\boldsymbol{\theta}( \mathbf{y}) \big) +D_\mathbf{w}\big(G_\boldsymbol{\theta}( \mathbf{y}) \big) \Big)
> >             \Big]  \\
> >         =
> > $$
> >
> > $$
> >       \max_{ \mathbf{v}, \mathbf{w}} \min_{\boldsymbol{\theta}, p}
> >            \mathbb{E}_\text{$\mathbf{y} \sim real$} \Big[ \langle \mathbf{v}, \mathbf{y} \rangle +\langle \mathbf{w}, [y_i^2] \rangle \Big]
> >           - \\  \mathbb{E}_\text{$ \mathbf{z} \sim \mathcal{N}(0,\mathbf{I} ), \zeta \sim \mathcal{N}(0,1) $}
> >             \Big[
> >                 (p + \zeta)\cdot
> >                 \Big( \langle \mathbf{v}, - (\mathbf{z} + \boldsymbol{\theta} ) \rangle +
> >                     \langle \mathbf{w}, [(\theta_i + z_i)^2] \rangle  \Big)
> >                  \\
> >           +  \big(1- (p + \zeta) \big) \cdot \Big( \langle  \mathbf{v}, - (\mathbf{z} + \boldsymbol{\theta} ) \rangle +
> >                     \langle \mathbf{w}, [\big( - (\theta_i + z_i) \big)^2] \rangle  \Big) \Big]  \\
> >         =
> > $$
> >
> > $$
> >           \max_{ \mathbf{v}, \mathbf{w}} \min_{\boldsymbol{\theta}, p}
> >            \mathbb{E}_\text{$\mathbf{y} \sim real$} \Big[ \langle \mathbf{v}, \mathbf{y} \rangle \Big] + \mathbb{E}_\text{$\mathbf{y} \sim real$} \Big[\langle \mathbf{w}, [y_i^2] \rangle \Big]
> >           -  \mathbb{E}_\text{$ \mathbf{z} \sim \mathcal{N}(0,\mathbf{I} ), \zeta \sim \mathcal{N}(0,1) $}
> >             \Big[
> >                 (p + \zeta)\cdot
> >                 \Big( \langle \mathbf{v}, \mathbf{z} + \boldsymbol{\theta} \rangle +
> >                     \langle \mathbf{w}, [(\theta_i + z_i)^2] \rangle  \Big)
> >                 +  \\
> >               \big(1- (p + \zeta) \big) \cdot \Big( \langle \mathbf{v}, -(\mathbf{z} + \boldsymbol{\theta} ) \rangle +
> >                     \langle \mathbf{w}, [(\theta_i + z_i)^2] \rangle  \Big) \Big] \\
> >         =
> > $$
> >
> > $$
> >       \max_{ \mathbf{v}, \mathbf{w}} \min_{\boldsymbol{\theta}, p}
> >             \Big \langle \mathbf{v}, \mathbb{E}_\text{$\mathbf{y} \sim real$} \Big[ \mathbf{y} \Big] \Big \rangle + \Big\langle  \mathbf{w}, \mathbb{E}_\text{$\mathbf{y} \sim real$} \Big[ [y_i^2] \Big] \Big\rangle
> >           -
> >                 p \cdot
> >                 \Big( \Big\langle \mathbf{v},
> >                         \mathbb{E}_\text{$ z \sim \mathcal{N}(0,\mathbf{I})$ }\Big[ (\mathbf{z} + \boldsymbol{\theta} ) \Big] \Big\rangle +
> >                     \Big\langle \mathbf{w}, \mathbb{E}_\text{$ z \sim \mathcal{N}(0,\mathbf{I})$ } \Big[ [(\theta_i + z_i)^2] \Big] \Big\rangle \Big) \\
> >                 -  \big(1- p \big) \Big( \cdot \Big\langle \mathbf{v},  \mathbb{E}_\text{$ z \sim \mathcal{N}(0,\mathbf{I})$ }  \Big[  - (\mathbf{z} + \boldsymbol{\theta} ) \Big] \Big\rangle +
> >                     \Big\langle \mathbf{w},  \mathbb{E}_\text{$ z \sim \mathcal{N}(0,\mathbf{I})$ } \Big[ [(\theta_i + z_i)^2] \Big] \Big \rangle   \Big) \\
> >         = \\
> > $$
> >
> > $$
> >       \max_{ \mathbf{v}, \mathbf{w}} \min_{\boldsymbol{\theta}, p}
> >             \Big \langle \mathbf{v}, \Big( \pi_1 \boldsymbol{\mu} + \pi_2 (- \boldsymbol{\mu}) \Big) \Big\rangle + \Big\langle  \mathbf{w}, [1 + \mu_i^2] \Big\rangle
> >           - \\
> >                 p \cdot
> >                 \Big( \Big\langle \mathbf{v},
> >                       \boldsymbol{\theta} \Big\rangle +
> >                     \Big\langle \mathbf{w}, [(\theta_i^2 + 1)] \Big] \Big\rangle \Big)
> >                 - \big(1- p \big) \cdot  \Big(  \Big\langle \mathbf{v},  - \boldsymbol{\theta} \Big\rangle +
> >                     \Big\langle \mathbf{w},   [(\theta_i^2 + 1)] \Big \rangle   \Big) = \\
> > $$
> >
> >
> > $$
> > \max_{ \mathbf{v}, \mathbf{w}} \min_{\boldsymbol{\theta}, p}
> >     (\pi_1 - \pi_2) \mathbf{v}^T \boldsymbol{\mu} - 2p \mathbf{v}^T \boldsymbol{\theta} + \mathbf{v}^T \boldsymbol{\theta} + \sum_i w_i( \mu_i^2- \theta_i^2)
> > $$

---

> > ### Author Response · Authors · 2021-11-23
> > **Response to Reviewer FpwP (3/5)**
> >
> > ## Calculations
> > By the definition of Variance, we get that :
> > $\mathrm{Var} \big[ x_i \big] = \mathbb{E}[{x_i^2}] - \big( \mathbb{E}[{x_i}] \big)^2 \Leftrightarrow \mathbb{E}[{x_i^2}] = \mathrm{Var} \big[ x_i \big] + \big( \mathbb{E}[{x_i}] \big)^2$. More precisely for any $\vec{x} \sim \mathcal{N}{(\mu, I)}$, we get $\mathbb{E}[{x_i^2}] = 1 +  \mu_i ^2$.
> >
> > Additionally, after calculations, we can get that:
> >
> > Moreover, it is easy to check that for a mixture $\mathcal{D}(\mu_1, \sigma_1\mathbf{I},\mu_2, \sigma_2\mathbf{I},
> >     \pi_1,\pi_2)$ of two multi-dimensional normal distributions $\mathcal{N}(\mu_1, \sigma_1\mathbf{I}), \mathcal{N}(\mu_1, \sigma_1 \mathbf{I})$ with corresponding weights $\pi_1, \pi_2$:
> > $$
> > \mathrm{Var}_\text{$x\sim \mathcal{D}$}[x_i] = \pi_1  \sigma_1^2 + \pi_2 \sigma_2^2 + \pi_1 \mu_\text{$1,i$}^2 +  \pi_2 \mu_\text{$2,i$}^2 - ( \pi_1 \mu_\text{$1,i$} + \pi_2 \mu_\text{$2,i$} )^2
> > $$
> >
> > Specifically for the case of $\mu_1 = \mu = - \mu_2$, this equivalent with:
> > $$
> >     \mathrm{Var}[x_i] = 1 + \mu_i^2  - ( \pi_1 \mu_{i} - \pi_2 \mu_{i} )^2
> > $$
> > Hence:
> > $$
> >     \mathbb{E} [x_i^2] = \mathrm{Var}[x_i] + ( \pi_1 \mu_{i} - \pi_2 \mu_{i} )^2 = 1 + \mu_i^2
> > $$
> >
> > Specifically for the case of $\mu_1 = \mu = - \mu_2$, this equivalent with:
> > $$
> >         \mathrm{Var}[x_i] = 1 + \mu_i^2  - ( \pi_1 \mu_{i} - \pi_2 \mu_{i} )^2
> > $$
> >
> > Hence:
> >     \begin{align*}
> >         \mathbb{E} [x_i^2] = \mathrm{Var}[x_i] + ( \pi_1 \mu_{i} - \pi_2 \mu_{i} )^2 = 1 + \mu_i^2
> >     \end{align*}

---

> > ### Author Response · Authors · 2021-11-23
> > **Response to Reviewer FpwP (2/5)**
> >
> > # Questions
> >
> > Projected gradient Descent-Ascent (Projected GDA), (more specifically, simultaneous Projected Gradient Descent-Ascent (simGDA) ) is an algorithm that is utilized in order to find Nash Equilibria of a given utility function $f$, for a minimizing agent $x$ and a maximizing player $y$.
> >
> > Much like in an elementary minimization problem, one would use gradient descent to reach a minimum of the objective function, in a min-max optimization setting, one agent tries to minimize the function with respect to the variables he/she controls ($x$) moving them towards the direction $f$ is minimized ($-\nabla_x f$: antiparallely to the gradient wrt to $x$). The maximizing agent moves their variables $y$ to the direction that the function $f$ should be maximized (i.e. $\nabla_y f$:parallely to the gradient wrt to $y$) Hence, gradient  *ascent*. Since these steps take place simultaneously we refer to this process as gradient descent-ascent.
> >
> >
> > When both updates are realized, there is no guarantee that the new values of $x, y$ should stay in the set of feasible solutions, hence we utilize the projection operator $ \Pi_\text{$\mathcal{X}_i $}, \Pi_\text{$\mathcal{Y}_i $} $ . As such, the new values get projected to a new value inside the feasible set.
> >
> > ## Explanation of the notation of the algorithms
> > In the case of the extragradient method, which constitutes a variant of GDA, instead of immediately calculating a gradient for each variable, an extra, intermediate step is made in order to calculate an (extra) gradient to make the final update. *$k+1/2$* stands for the intermediate step made that is only used for the calculation of the (extra) gradient. This method has been shown to provide with last-iterate convergence to Nash Equilibria in convex-concave zero-sum games as opposed to average convergence that simple GDA provides. ( The projection operators have exactly the same role as in the simple projected GDA algorithm.)
> >
> > The optimistic gradient descent-ascent (OGDA) is yet another variant of GDA. It utilizes a concept of optimism by assuming that the next step can be partially anticipated by the gradient computed at the previous step ($ x_{k-1}, y_{k-1} )$.  It can be thought of as a negative momentum variant of GDA, as well. OGDA also enjoys last-iterate convergence to Nash Equlibria in two-person convex-concave zero-sum games.
> >
> > We note that last-iterate convergence is something sought after especially in applications such as GANs in which averaging the parameters of previous states is prohibitive due to their enormous size.
> >
> > We dismissed further explanation from our paper as these algorithms are ubiquitous in the relevant min-max optimization literature.

---

> > ### Author Response · Authors · 2021-11-23
> > **Response to Reviewer FpwP (1/5)**
> >
> > We thank the reviewer for their time and remarks. Unfortunately, there seem to be some misunderstandings and misinterpretations regarding our paper’s contributions, which we hope to clarify in our point-to-point replies below.
> >
> > # Novelty
> > The reviewer highly criticized our work stating that "This paper cannot make us understand two-team zero-sum games better."
> >
> > To argue for such a dismissal, he/she claimed that since computing a Nash equilibrium in N-player games is PPAD-complete (Daskalakis et al., 2009), **it seems more than obvious that two-team zero-sum games** is CLS-hard (i.e., it is unlikely to have a polynomial-time algorithm for computing a Nash equilibrium). ''
> >
> > Firstly, we would like to remind the reviewer that a computational hardness result for a family of games (or problems for that matter) does not preclude potentially efficient results for subfamilies of games \& dynamics.
> >
> > In other words, hardness/impossibility results carry over to the most general collections of a problem and not to subfamilies that adhere to certain restrictions/relaxations. Actually, providing hardness results for more concrete sub-domains of the general problem enhances our understanding of the difficulty of the task (much like we attempt to do in our work).
> >
> > Team zero-sum games seem to belong in the middle of the complexity taxonomy since the simple case of a **Two-Player Zero-Sum Game** is polynomially tractable while the case of a **Two-Team with a large number of players Zero-Sum Game** seems to be closer to Coordination/Potential games whose complexity (PLS/CCLS) is likely to be easier than the general PPAD-hardness.
> >
> > A great example is the case of "Two-Player Zero-Sum Games" which are solvable via LP/Interior Point Methods, etc (due to duality). Another example is the case of the “Decomposed Polymatrix Zero-Sum N-player games", where similar techniques also apply.
> >
> > In our opinion, the correct interpretation of this result should be that "Team Zero-Sum Games are critically \& provably more computationally difficult" than the case of two players yet there exist local search methods that provably provide solutions for certain instances.
> >
> > Going beyond the complexity result, the same rationale holds for the referenced publication for the failures of game dynamics. The impossibility results propagate only to a harder family of games (upwards with respect to hardness, not downwards). Otherwise, the contemporary line of research in Games [1-5] which includes some celebrated and highly-cited new results should be dismissed as mere "incremental contribution[s]".

---

### Official Review · Reviewer_iJde · 2021-11-01

**Correctness:** 3
**Technical Novelty And Significance:** 2
**Empirical Novelty And Significance:** 2
**Recommendation:** 6
**Confidence:** 4

**Main Review:**

Strengths:
1. The setting of two-team games is important, even the zero-sum version has many applications.
2. The techniques used in the paper are well-motivated and, as far as I checked, are correct.

Weaknesses:
1. The detailed proofs are relatively simple. Although not trivial, most conclusions are not suprising. The reduction for CLS-hardness is somewhat straightforward. The locally convergence of the proposed KPV-GDA is proved by showing the existance of suitable K and P but no construction is provided.
2. There are several issues related to the presentation. The general organization is fine but there are many typos and misused capital chatractors. The most important one is the ambiguity in Section 4 (last two sentences). Basically, I cannot understand the settings and the results provided in the experiments, although this does not influence the theoretical parts.
3. One minor concern about the illustraive exmample on multi-agent GANs: The Nash equilibrium in this paper is defined in the sense of per player, i.e. each individual player cannot improve itself by unilaterally changing its strategy. However, it seems MGANs should in fact care about the equilibrium where each team cannot improve itself by unilaterally changing all its team members' strategies (still restricted in the space of Cartesian product of simplices). The later, just called per team NE as I have not the official name, is defined similarly as the team max-min equilibrium except both teams can have multiple players. Each per team NE, although not necessarily exists, must be a per player NE, but not vise versa.

**Summary Of The Paper:**

This paper studies the equilibrium compution in two-team zero-sum games. Finding the per player Nash is proved to be CLS-hard, and many popular gradient-based algorithms are proved to be not stable. A vaned gradient descent ascent algorithm is proposed to address the instability, shown to locally converge.

**Summary Of The Review:**

I think this paper is blow the bar of acceptance. Currently I am negative on it due the weaknesses I mentioned, but may change to be positive if those issues can be addressed.

---

> ### Author Response · Authors · 2021-11-23
> **Response to Reviewer iJde**
>
> We thank the reviewer for their time and insightful comments.
>
> 1. The initial version did only provide an existential proof for matrices $K,P$. We supplemented the revised version with a sufficient condition such that there exists a simple parametrization for the matrices in question. We refer to the response we provided to Reviewer 89w7: https://openreview.net/forum?id=UyBxDoukIB&noteId=3Mq_MuasswM.
>
> 2. You are right regarding the presentation. We did our best to improve it. With regards to the experiments in Section 4:The experiments were conducted with a toy dataset of a uniform of 8 gaussian functions each with a covariance matrix equal to $0.03\cdot \mathbf{I}$ and mean values located at eight equidistant points upon the circle centered at the origin with a radius equal to $2$. \
> Upon each iteration, $12$ rounds of updates take place for the discriminators. Batches of $250$ samples from each generator are fed to every discriminator along $120$ datapoints from the real distribution. Next, every generator is trained for $1$ round. Each generates a batch of $250$ datapoints upon whose loss it is trained. \
> Similarly, for the case of single big generator vs the single big discriminator, in every iteration $12$ rounds of updates are realized in order to train the discriminator and $1$ round in order to train the generator. The batch sizes are $200$ real datapoints and $200$ generated.
>
> 3. In the case of the illustrative example every agent is trained independently. Were they to be trained as a coordinated team, the setting would not defer from a conventional GAN. The fixed points of the objective function have a statistical meaning when every agent is considered independent. That is, the distribution is learned when a per-player Nash Equilibrium is reached.
>
> With kind regards,\
> The authors

---

> > ### Comment · Reviewer_iJde · 2021-11-23
> > **RE**
> >
> > Thanks for your clarification. It addressed most of my concerns. However, as you mentioned, the experiments part is still kind of simple.

---

### Official Review · Reviewer_rW5j · 2021-11-03

**Correctness:** 3
**Technical Novelty And Significance:** 2
**Empirical Novelty And Significance:** 2
**Recommendation:** 6
**Confidence:** 3

**Main Review:**

Overall, I am a bit conflicted for this paper. I appreciate the CLS-hardness result, and like the idea of using control theory to stabilize the dynamics of GDA. At the same time, I think the paper is lacking in many aspects:

#1. First of all, I was surprised to see no mention of the dichotomy between team maxmin equilibrium [3] (which to my understanding is what the authors are focusing on), versus its convexification TMECor [1][2] until the appendix. TMECor allows for correlated strategies to be used within the team. As independent strategies are a special case of correlated strategies, TMECor always allows for higher social welfare compared to TME against the same opponent. Also, the use of correlation is not an obstacle, as long as the team members can agree on a common shared signal to use to seed the same random number generator (for example, the team member having access to the current time would often be good enough).

But above all, beyond higher welfare and limited additional assumptions, TMECor is a convex concave problem. So, I think that TME makes for a pretty non compelling solution concept compared to TMECor. Showing hardness of TME is definitely interesting, but I believe that a robust discussion about TMECor and its benefits should be present in the paper. Realistically, it's hard for me to imagine a practical case where I'd rather use TME than TMECor.

#2. The paper is a bit hard to read. I've numbered here a few questions that would help clarify a few of the obscure points in the rebuttal:

a) It was not immediately clear to me what "converge *locally* to a point" means

b) Is KPV-GDA defined in the introduction the same as GDA-KPV mentioned in Theorem 3.7?

c) In the conclusions you mention that your stabilized algorithm "manages to stabilize *around* Nash equilibria". What does around mean here? In Theorem 3.7 you talk about "local convergence" to a Nash equilibrium. So, it that the same thing?

d) What can be said about the matrices K and P in Theorem 3.7? Does it mean that the theorem is purely an existential result? Or is there an algorithm to compute the required matrices?

#3. I think the paper would benefit from polishing the language. That goes beyond typos: I think the overall language of the paper needs to be made more precise, as it often came off to me as vague and handwavy. For example, beyond all examples mentioned above in point 2.:
- In the abstract, you write "We present a family of team games whose induced utility is non-multilinear with non-attracting per-se mixed Nash equilibria". What does "per-se" bind to here? The term is only used once, and it was not clear to me if it was a technical term I was not familiar with (a "per-se Nash equilibrium"), or if it instead referred to "per-se not-attracting", which again was not clear. Is the point attracting, or not attracting, or something else that needs to be explained? Also, if "per-se" referred to the Latin expression, I believe it does not need a hyphen.
- In the abstract again, you write ".. is not possible using .. (GDA), its optimistic variant and extra gradient". I think it would have been clearer to say "its extragradient variant".
- The paper is inconsistent between "two-player zero-sum" and "two player zero-sum".
- In the introduction, you talk about a certain min-max approach missing "the critical component of the collective strategy making". What is the critical component? I wish you had been way more specific.
- Introduction: "computing local Nash Equilibrium .. are" -> is
- Introduction: "is computationally harder of finding" -> than
- Introduction: "From optimization perspective" -> "From an"
- I encourage the authors to improve the figures, which are hard to read due to the fonts (very low contrast, the text looks pale instead of black) and font sizes used.
- Section 3.4: "our machiner" -> "machinery"
- Experiments: "Iter 3000" refers to data generated by 3000 episodes worth of training? It would be great to dive a more complete description as to how the experiments were run in the Experimental section of the paper.


[1] Basilico et al. "Team–Maxmin Equilibrium: Efficiency Bounds and Algorithms"

[2] Celli & Gatti "Computational Results for Extensive-Form Adversarial Team Games".

[3] von Stengel & Koller "Team maxmin equilibria".

**Summary Of The Paper:**

The main contributions of the paper are as follows. First, the authors show show that the computation of Nash equilibrium in two-team zero-sum game is CLS-hard. As a result, GDA and its variants (including optimistic GDA and extragradient) cannot---in general---be used to converge to the Nash equilibrium. Then, the authors propose a "stabilized" version of GDA (called KPV-GDA) obtained through certain stabilization techniques in control theory.

**Summary Of The Review:**

I appreciated the theoretical results of the paper, though I believe more needs to be done to justify the model. I also think the writing could be significantly improved. Overall, I am somewhat on the fence. I welcome a thorough discussion with the authors.

---

> ### Author Response · Authors · 2021-11-23
> **Response to Reviewer rW5j (3/3)**
>
> ## Clarification about terminology
> We would like to thank the reviewer for the opportunity to clarify this subtle issue. Indeed, a typical convention in ML Optimization community is to exchange freely the terms of *local convergence, stabilization*.
>
> Topologically speaking the crucial question is the morphology of the stable manifold set of a fixed point.
>
> More precisely,
> let $x^∗$ be a fixed point (Our MNE) of a local diffeomorphism $g$ (Our game dynamics). Additionally, let $E_s$ be the span of the eigenvectors of $Dg(x^∗)$ corresponding to eigenvalues of magnitude less than or equal to one. Then, the *Stable Manifold Theorem* states that
> there is an embedded disk $W_cs^loc$ tangent to $E_s$ at $x^∗$ called the local stable center manifold. Moreover, there is a neighborhood $B$ of $x^∗$, such that $g(W_{cs}^{loc})\cap B \subset W_{cs}^{loc}, and \cap_{k=1}^{\infty}g^{-k}(B) \subset W_{cs}^{loc}.$
>
> Therefore, whenever the Jacobian of a method includes eigenvalues greater than one, $E_s$ vector space and the stable manifold around the mixed Nash equilibrium is not of full dimension. Therefore, only a measure zero initialization can converge to this point.
>
> On the other hand, if the spectral norm of a method's Jacobian is less than one then the corresponding MNE has a non-trivial stable manifold of full dimension. Equivalently, the corresponding equilibrium is Lyapunov stable.
>
> ## Clarification for experiment
> The experiments were conducted with a toy dataset of a uniform of 8 gaussian functions each with a covariance matrix equal to $0.03\cdot \mathbf{I}$ and mean values located at eight equidistant points upon the circle centered at the origin with a radius equal to $2$.
>
> Upon each iteration, $12$ rounds of updates take place for the discriminators. Batches of $250$ samples from each generator are fed to every discriminator along $120$ datapoints from the real distribution. Next, every generator is trained for $1$ round. Each generates a batch of $250$ datapoints upon whose loss it is trained.
>
> Similarly, for the case of a single big generator vs a single big discriminator, in every iteration $12$ rounds of updates are realized in order to train the discriminator and $1$ round in order to train the generator. The batch sizes are $200$ real datapoints and $200$ generated.
>
> Thank you very much for your comments again.
>
> Kind regards,\
> *The Authors*
>
>
> [1] Tim Roughgarden. Intrinsic robustness of the price of anarchy. In Proceedings of the forty-first
> annual ACM symposium on Theory of computing, pp. 513–522, 2009
>
> [2] Youzhi Zhang, Bo An, and Jakub Cerny. Computing ex ante coordinated team-maxmin equilibria
> in zero-sum multiplayer extensive-form games. arXiv preprint arXiv:2009.12629, 2020.
>
> [3] Joel Z Leibo, Vinicius Zambaldi, Marc Lanctot, Janusz Marecki, and Thore Graepel. Multi-agent
> reinforcement learning in sequential social dilemmas. arXiv preprint arXiv:1702.03037, 2017
>
> [4] Yongcan Cao, Wenwu Yu, Wei Ren, and Guanrong Chen. An overview of recent progress in the
> study of distributed multi-agent coordination. IEEE Transactions on Industrial informatics, 9(1):
> 427–438, 2012.
>
> [5] Hassam Ullah Sheikh, Mina Razghandi, and Ladislau Boloni. Learning distributed cooperative
> policies for security games via deep reinforcement learning. In 2019 IEEE 43rd Annual Computer
> Software and Applications Conference (COMPSAC), volume 1, pp. 489–494. IEEE, 2019.

---

> ### Author Response · Authors · 2021-11-23
> **Response to Reviewer rW5j (2/3)**
>
> Although the initial version of the paper held only an existential proof, in the revised one we added a sufficient condition under which matrices $K,P$ can be chosen with a very simple structure (multiples of the identity matrix)
>
> For further discussion we refer to the comment to reviewer 89w7:
> https://openreview.net/forum?id=UyBxDoukIB&noteId=3Mq_MuasswM
>
> (Yes, KPV-GDA is the same as GDA-KPV. We polished the language and the general text.)

---

> ### Author Response · Authors · 2021-11-23
> **Response to Reviewer rW5j (1/3)**
>
> We thank the reviewer for the valuable comments. We will first answer your individual questions.
>
> We would like to apologize for the typos. We already took care of them in the revised version that we uploaded, we correct the mentioned expressions and the various issues in annotations.
>
> ## TMECor
> We would firstly like to mention that the discussion of related work was regretfully chosen to be included in the Appendix because of the page limit. In the case of acceptance, when the page limit is relaxed, we intend to transfer part of this discussion into the main text.
>
> Below, we would like to stress some differences between the different equilibrium notions (*TMECor, TME, and per-player NE*) explaining the game-theoretic rationalism of our choice.
>
> Initially, it is easy to see that TMECor similarly with the Coarse correlated equilibria can achieve better social welfare but as in the case of congestion games, found in literature, their price of anarchy can become significantly worse [1].
> In order to give a better result, TMECor requests an a priori knowledge of the game. For example, as [2] mentioned:
>
> `For example, in multiplayer poker games, a team may play against an adversary player, but they cannot communicate and discuss their strategy during the game due to the rule.`
>
> However in many nowadays challenging AI models, the agents/players of each team do not necessarily have knowledge of the actual game. Thus, the team players can only decide in advance only about their own dynamics. Thus, the main conceptual reason that we focus on NE per player in comparison with the aforementioned TME relaxations is to study the interplay between the individual and team incentives in competitive tasks. It is also a way of reasoning about games in which players do not necessarily need to know in which team they fall into in advance.
>
> Problems of this kind are prevalent in modern ML applications like strategic conflict resolution  [3],  coordination between autonomous vehicles  [4] , or collaboration of agents in defensive escort teams  [5]. In this kind of games, each agent is simultaneously working towards maximizing its own payoff  *(local reward)* as well as the collective success of the team *(global reward)*.
>
> As it is stated again in [2],
>
> `Celli and Gatti (2018) show that the "ex ante" coordination can be modeled using a coordination device, assuming that the adversary does not observe any signal from the device. The team members agree on a planned strategy (e.g., a mixed strategy) in the planning phase, and then, just before the game starts, the coordination device randomly picks a pure joint strategy (from the planned strategy) for the team members to act upon.`
>
> Therefore,  this TMECor interpretation would critically correspond to a high-level two-*metaplayer* zero-sum game, where the metaplayer is the team as a whole entity with every individual player in perfect coordination with its teammates.
>
> Finally, an additional reason, the notion of "per player Nash equilibria" can be seen as a smoother figure of merit for the performance of a defensive team against an adversary. Notably, in the vast majority of the aforementioned literature, a single superpowerful adversary has been used. However, using a single-agent adversary can describe a universally fair model.By contrast, the notion of per player NE expresses the  model hurdles of both simultaneous *team* intra-collaboration and inter-competition.

---

> > ### Comment · Reviewer_rW5j · 2021-11-30
> > **Thanks for the response**
> >
> > Thanks for the response and for the stimulating discussion around TME and TMECor.
> >
> > * "By contrast, the notion of per player NE expresses the model hurdles of both simultaneous team intra-collaboration and inter-competition".  I am not sure I got what you meant with this. TMECor definitely has those components too. In fact, TMECor can be seen as the convexification of TME [1].
> >
> > * In fact, a major game-theoretic weakness of TME is the fact that the minmax theorem fails to hold. That makes TME very weak: unlike two-player zero-sum Nash equilibrium, in a two-team zero-sum TME swapping the max and the min results in different values. The same does not happen in TMECor. So, I am still very much not convinced that TME is superior to TMECor, and I would want to see a careful discussion in the paper explaining that TME has significant drawbacks compared to TMECor. As you point out, it also has some advantages. It is up to the reader to figure out whether TME or TMECor is more appealing in their setting of interest, but for the final version I believe it is important to highlight the many drawbacks of TME (including lower social welfare, lack of convex concave structure, failure of min max theorem) compared to TMECor.
> >
> > That being said, I realize that the discussion between TME and TMECor is not the core of the paper. Since the authors have significantly improved the quality of the prose, I am more positively inclined towards this submission. I trust the authors to revise and expand the treatment of TMECor as discussed,
> >
> >
> > [1] G. Farina, A. Celli, N. Gatti and T. Sandholm. Ex Ante Coordination and Collusion in Zero-Sum Multi-Player Extensive-Form Games. NeurIPS 2018.

---

> ### Author Response · Authors · 2021-11-26
> **Discussion period soon ends. Further questions?**
>
> Given that the discussion period ends soon, we wanted to make sure that reviewers do not have any lingering questions. Let us know if we can help clarify something. We would like your feedback after our further clarifications and revision.

---

### Official Review · Reviewer_89w7 · 2021-11-04

**Correctness:** 4
**Technical Novelty And Significance:** 3
**Empirical Novelty And Significance:** Not applicable
**Recommendation:** 6
**Confidence:** 4

**Main Review:**

I have read the authors answer' and the other reviews. I am satisfied with their answers and do not have any concern other than the lack of clarity at some points. I am thus still in favour of its acceptance.
-----------------------------------------
I globally find this work very interesting, well writtend and I think the problem of two-team zero-sum games deserves careful considerations. My main concern is about the comparison with related work, which seems weird/incomplete. When reading the related paper from Daskalakis and Panageas for example, they already claim that GDA and OGDA may not converge to NE in zero sum-games (in last iterate). A main part of this work is a similar claim in two-team zero-sum games (Thm 3.5), but it trivially holds if it is already the case in classical zero-sum games. As a consequence, I think the better should better focus on comparing to average iterates of these algorithms, that are known to converge to equilibria, or to methods that are known to converge in last iterate, such as OMWU for example. The case of average iterates is only mentioned in Figure 4 and Remark 3.6, but having a longer discussion (+ a theoretical result) on this would be great in my opinion. For example, Figure 2 makes less sense as soon as we are considering average iterates.

The other main "weakness" of this paper is the fact that Theorem 3.7 only holds under specific choices of the matrices K and P, that depend on the problem parameters. This is mentioned by the authors in the conclusion, and I agree with them that this can be left as future work.

Minor comments:
- p.2: "is computationally harder of finding pure NE in a congestion game". Isn't it a typo? "than" instead of "of"?
- typos:
           p.3 "introduces" -> "introduced"
                 "their" -> "they are"
           p.5 "captures" -> "capture"
           p.6 "non-trivial", don't you mean "trivial"?
           p.9: "small number of neurons achieves to", the end of the sentence is missing

- could you please explain the meaning of equation (2.4) with words?
- the figures are lacking legends and are thus hard to interpret for some of them
- Theorem 3.1: I think it would be nice to define CLS hard before
- Theorem 3.7: what is exactly $\nabla_{x,x}$?

**Summary Of The Paper:**

This work considers two-team zero-sum games where two teams with opposite objectives are facing each other. While the literature is quite extensive on learning in zero-sum games, very little is known for two-team zero sum-games. This paper first shows that this problem is much harder than classical zero-sum games by showing that the computation of a NE is CLS hard. Besides showing the failure of classical optimization methods for zero-sum games, the authors propose an optimisation algorithm better suited to this setting, which converges locally to a NE for carefully tuned hyperparameters.

**Summary Of The Review:**

This paper is overall interesting and significant for the problem of two team zero sum games. My main concern is how it relates to previous works and I hope the authors could clarify this point.

---

> ### Author Response · Authors · 2021-11-23
> **Response to Reviewer 89w7 (3/3)**
>
> [1] Wei, C.Y., Lee, C.W., Zhang, M. and Luo, H., 2020, September. Linear Last-iterate Convergence in Constrained Saddle-point Optimization. In International Conference on Learning Representations.
>
> [2] Panayotis Mertikopoulos, Bruno Lecouat, Houssam Zenati, Chuan-Sheng Foo,
> Vijay Chandrasekhar, and Georgios Piliouras. Optimistic mirror descent in
> saddle-point problems: Going the extra(-gradient) mile. In ICLR, 2019
>
> [3] Vinyals, O., Babuschkin, I., Czarnecki, W.M. et al. Grandmaster level in StarCraft II using multi-agent reinforcement learning. Nature 575, 350–354 (2019). https://doi.org/10.1038/s41586-019-1724-z
>
> [4] James Hannan. Approximation to Bayes risk in repeated play. In Melvin Dresher, Albert William
> Tucker, and P. Wolfe (eds.), Contributions to the Theory of Games, Volume III, volume 39 of Annals
> of Mathematics Studies, pp. 97–139. Princeton University Press, Princeton, NJ, 1957.
>
> [5] Sergiu Hart and Andreu Mas-Colell. A simple adaptive procedure leading to correlated equilibrium.
> Econometrica, 68(5):1127–1150, September 2000.
>
> [6] Noam Nisan, Tim Roughgarden, Éva Tardos, and V. V. Vazirani (eds.). Algorithmic Game Theory.
> Cambridge University Press, 2007.
>
> [7] Eddie Dekel and Drew Fudenberg. Rational behavior with payoff uncertainty. Journal of Economic
> Theory, 52:243–267, 1990
>
> [8] Drew Fudenberg and Jean Tirole. Game Theory. The MIT Press, 1991.
>
> [9] Yannick Viossat and Andriy Zapechelnyuk. No-regret dynamics and fictitious play. Journal of
> Economic Theory, 148(2):825–842, March 2013.
>
> [10] Fearnley, J., Goldberg, P.W., Hollender, A. and Savani, R., 2021, June. The complexity of gradient descent: CLS= PPAD∩ PLS. In Proceedings of the 53rd Annual ACM SIGACT Symposium on Theory of Computing (pp. 46-59).
>
> [11] Christos H. Papadimitriou. The complexity of the Lin-Kernighan heuristic for the
> traveling salesman problem. SIAM Journal on Computing, 21(3):450–465, 1992.
> doi:10.1137/0221030.
>
> [12] Alex Fabrikant, Christos Papadimitriou, and Kunal Talwar. The complexity of pure Nash
> equilibria. In Proceedings of the 36th ACM Symposium on Theory of Computing (STOC),
> pages 604–612, 2004. doi:10.1145/1007352.1007445.
>
> [13] Constantinos Daskalakis and Christos Papadimitriou. Continuous local search. In Pro-
> ceedings of the 22nd ACM-SIAM Symposium on Discrete Algorithms (SODA), pages
> 790–804, 2011. doi:10.1137/1.9781611973082.62.

---

> ### Author Response · Authors · 2021-11-23
> **Response to Reviewer 89w7 (2/3)**
>
> ## Minor Concerns
> * No, we meant to say *non-trivial*
> * fixed the sentence
> * We apologize for the quality of the figures, we did our best to improve the quality in the revised version of the draft
> * As a recent result[10] shows CLS is equal to the intersection to PLS and PPAD, two important classes of total problems. PPAD captures diverse problems in combinatorics and (non-)cooperative game theory, like the $\varepsilon$-approximation of a mixed Nash Equilibrium in a graphical game or the computation of market equilibria. PLS, for “Polynomial Local Search”, captures problems of finding a local minimum of an objective function f, in contexts where any candidate solution x has a local neighbourhood within which we can readily check for the existence of some other point having a lower value of f. Many diverse local optimization problems have been shown complete for PLS, attesting to its importance. Examples include searching for a local optimum of the TSP according to the Lin-Kernighan heuristic [11], and finding pure Nash equilibria in many-player congestion games [12]. The complexity class CLS (“Continuous Local Search”) was introduced by Daskalakis and Papadimitriou [13] to classify various important problems that lie in both PPAD and PLS. CLS is seen as a strong candidate for capturing the complexity of some of those important problems, like the general versions of Banach’s fixed point theorem, computation of KKT points, computation of gradient descent fixed points etc.
>
> * In Figure 2 the main observation is that algorithms that generally succeed in a two-player game (like OGDA) cycle around while KPV-GDA converges to the Nash Equilibrium in the last iterate.
>
> * In contrast, in figure 4, we show that even if we examine the time average $\dfrac{1}{T}\sum_{t\in[T]} x_t$ convergence of GDA/OGDA/ExtraGradient and our method. Since our method converges in last-iterate would converge also in the time-average sense to the Mixed Nash Equilibrium. On the contrary, the gradient methods converge to a different CCE point.
>
> * We overload the notation of $\nabla_\text{$x, y$}$ to mean the submatrix of the Hessian: $ \begin{bmatrix}
>   \dfrac{\partial^2 f}{\partial x_1\, \partial y_1} & \dfrac{\partial^2 f}{\partial x_1\,\partial y_2} & \cdots & \dfrac{\partial^2 f}{\partial x_1\,\partial y_n} \\\\[2.2ex]
>   \dfrac{\partial^2 f}{\partial x_2\,\partial y_1} & \dfrac{\partial^2 f}{\partial x_2 \, \partial y_2} & \cdots & \dfrac{\partial^2 f}{\partial x_2\,\partial y_n} \\\\[2.2ex]
>   \vdots & \vdots & \ddots & \vdots \\\\[2.2ex]
>   \dfrac{\partial^2 f}{\partial x_n\,\partial y_1} & \dfrac{\partial^2 f}{\partial x_n\,\partial y_2} & \cdots & \dfrac{\partial^2 f}{\partial x_n\,\partial y_n}
> \end{bmatrix}$. In the case the reviewer finds it necessary, we will make notation clearer in the camera-ready version.
>
>
> ## KP Design
>
> In general, the problem of designing matrices $K, P$ is quite challenging. In our experiments we restricted ourselves to the case in which $K, P$ are both multiples of the identity matrix, i.e., $K = k \cdot \mathbf{I}$ and $P = p \cdot \mathbf{I} $. With such a restriction, the theorem we provided does not hold universally. Nevertheless, we provide a sufficient condition under which local convergence is guaranteed if parameters $p, k$ are selected carefully (see also the revised version of the paper):
>
> To be more specific, consider the following matrix $H$ at the Nash Equilibrium: $$ H = \left( \begin{array}{cc}
> -\nabla_{\mathbf{x}\mathbf{x}}U(\mathbf{x}^*,\mathbf{y}^*) & -\nabla_{\mathbf{x}\mathbf{y}}U(\mathbf{x}^*,\mathbf{y}^*)\\\\
> \nabla_{\mathbf{y}\mathbf{x}}U(\mathbf{x}^*,\mathbf{y}^*) &\nabla_{\mathbf{y}\mathbf{y}}U(\mathbf{x}^*,\mathbf{y}^*)
> \end{array}\right)$$
> and define $E$ as the set of its eigenvalues that have a positive real part. Moreover, let $\alpha, \beta$ be defined as:
>
> $$\beta=\min_\text{$\rho\in E$} \frac{\textrm{Re}(\rho)^2 +\textrm{Im}(\rho)^2}{\textrm{Re}(\rho)}>\max_\text{$\rho\in E$} \textrm{Re}(\rho)=\alpha. $$
>
>
> For any $k\in (-\beta,-\alpha)$ and $p, \eta$ sufficiently small, positive constants the KPV-GDA method with chosen $K = k\cdot \mathbf{I},P  = p \cdot \mathbf{I}$ converges locally to $(\mathbf{x}^*,\mathbf{y}^*)$.
>
> Thank you again for your detailed comments and your positive evaluation and we hope to higher support after our clarifications!
>
>
>
> Kind regards,\
> The authors

---

> ### Author Response · Authors · 2021-11-23
> **Response to Reviewer 89w7 (1/3)**
>
> Thank you very much for the detailed and thoughtful review, and for your positive evaluation and assessment. We reply to your precise questions below:
>
> ## Related Work
>
> The main concern of the reviewer was on how our work compares with the related work.
>
> The abundant number of different results that have appeared in the field can easily lead to misinterpretations. For example, the negative results of Daskalakis and Panageas refer to unconstrained min-max optimization, where even the existence of a Nash Equilibrium is not a given (notice that $\mathbb{R}^n$ is not bounded, hence the Brouwer/Kakutani/Nash fixed point theorems do not apply). By contrast, convergence results for Optimistic-GDA in games has been proved by the work of Chen-Yu Wei et.al [1] which was accepted in last year’s ICLR. Therefore, Optimistic-GDA in two-player zero-sum games, (min-max optimization over a probability simplex), **does** converge to Mixed Nash Equilibria.
>
> Thus, Optimistic-GDA and Extragradient seem to be excellent candidates for min-max optimization in two-team zero-sum games. Indeed, for the case of the Eulerian discretization of Gradient flow, aka *vanilla* gradient descent-ascent, we can already name results regarding the method’s cycling/divergent behavior. However, we believe that for the sake of completeness it would be useful to include it as the basic equation that was set as a stepping stone for much of the work of the min-max optimization community. One of the main contributions of our work is the construction of a family of games in which the state-of-art discretization, namely GDA, OGDA, ExtraGradient fail to even stabilize, let alone discover a Nash Equilibrium.
>
> ## The performance of OMWU and average-time convergence
> We would like to thank the reviewer for the interesting question about OMWU. We left out this method just for the sake of conciseness in the paper's narrative about gradient-based methods. In our revision, we have included in the appendix the corresponding failure result for the case of OMWU as well. More precisely, we show that OMWU will be repelled almost always from the Mixed Nash Equilibrium of the presented game.
>
> Together with the proof, we present an example of a two-team zero-sum game where OMWU clearly cycles around the equilibrium as shown in figure 2.
>
> In this point, we would like to remind the reader that the main scope of our work is the last-iterate convergence of our method since this is the critical requirement for contemporary applications in Machine Learning like GANs [2], or multi-agent learning like Starcraft's AI gaming tasks [3]. Intuitively, the reason is two-fold: 1. In a non-convex non-concave setting we would not expect that the averaging of the parameters of a GAN will output something meaningful 2. With regards to Reinforcement Learning and considering as an example the high-frequency trading regime, an average equilibrium is as bad as the variance of the expected losses, i.e. although we can expect to attain the value of the game in average, we could be led to bankruptcy long before the boom and bust cycle closes.
>
> A rather handwavy answer to this question is that “no-regret learning converges to an equilibrium in all games” [6], suggesting in this way that no-regret dynamics inherently gravitate towards game-theoretically meaningful states. However, at this level of abstraction, both the type of convergence as well as the specific notion of the equilibrium in question are not as strong as one would have hoped for. Formally, the only precise conclusion that can be drawn is the following: under a no-regret learning procedure, the empirical frequency of play converges to the game’s set of coarse correlated equilibria [4,5]. This leads to an important discrepancy with standard game-theoretic solution concepts on several grounds.
>
> First, even in 2-player games, coarse correlated equilibria may be exclusively supported on strictly dominated strategies [9], so they fail to address even the most basic requirement of rationalizability [7,8] Second, the paradigmatic game-theoretic solution concept is that of the Nash equilibrium (NE), and convergence to a Nash equilibrium is a much more tenuous affair.

---

### Author Response · Authors · 2021-11-23
**Summary of our revision**

We thank all reviewers for their valuable time and comments from which we learn a lot. We have made our first revision that addresses most of the major concerns. One notable change is that we improve our result for the $K,P$-Vaned GDA method. We strongly believe that this interplay of control theory machinery like washout-filters as well as the existence of such matrices $K$ and $P$ used to stabilize optimization algorithms are matters of independent interest.

In general, the problem of designing matrices $K, P$ is quite challenging. In our experiments we restricted ourselves to the case in which $K, P$ are both multiples of the identity matrix, i.e., $K = k \cdot \mathbf{I}$ and $P = p \cdot \mathbf{I} $. With such a restriction, the theorem we provided does not hold universally. Nevertheless, we provide a sufficient condition under which local convergence is guaranteed if parameters $p, k$ are selected carefully (see also the revised version of the paper):

To be more specific, consider the following matrix $H$ at the Nash Equilibrium: $$ H = \left( \begin{array}{cc}
-\nabla_{\mathbf{x}\mathbf{x}}U(\mathbf{x}^*,\mathbf{y}^*) & -\nabla_{\mathbf{x}\mathbf{y}}U(\mathbf{x}^*,\mathbf{y}^*)\\\\
\nabla_{\mathbf{y}\mathbf{x}}U(\mathbf{x}^*,\mathbf{y}^*) &\nabla_{\mathbf{y}\mathbf{y}}U(\mathbf{x}^*,\mathbf{y}^*)
\end{array}\right)$$
and define $E$ as the set of its eigenvalues that have a positive real part. Moreover, let $\alpha, \beta$ be defined as:

$$\beta=\min_\text{$\rho\in E$} \frac{\textrm{Re}(\rho)^2 +\textrm{Im}(\rho)^2}{\textrm{Re}(\rho)}>\max_\text{$\rho\in E$} \textrm{Re}(\rho)=\alpha. $$


For any $k\in (-\beta,-\alpha)$ and $p, \eta$ sufficiently small, positive constants the KPV-GDA method with chosen $K = k\cdot \mathbf{I},P  = p \cdot \mathbf{I}$ converges locally to $(\mathbf{x}^*,\mathbf{y}^*)$.


We strongly believe that exploring a different structure of matrices can give even tighter bounds in the design parameters of the method.
However, we believe that the optimal fine-tuning of all these constraints goes far beyond the scope of this paper.

Additionally, we provide provable instances of failure for the case of OMWU in two-team games after the insightful question by Reviewer 89w7.

Finally, we would like to apologize for the typos and the figures' low resolution and thank you for pointing them out. We did our best to correct them.

In more detail, our revision incorporates the following:

**Section 1 (Introduction)**:
Correction of typos/misspellings and a more precise refinement of previously ''vague'' statements.

**Section 2.1 (Preliminary Work)**:
In order not to violate the submission page limits, we set a remark for OMWU and we refer to the corresponding appendix for its definition.

**Section 2.2 (Illustrative Examples)**:
Improvement of images and correction of critical typos in the optimization of the M-Gans.


**Section 3 (Our Main Results)**:
The new results of this section include the failure results for OMWU and Selection Parameters for the $K,P$-Vaned GDA method.

Again in order not to violate the submission page limits, we defer the proofs to new sections in the supplementary material.


**Appendix A (Related Work)**:
We expand our discussion in the related work section to the recent work revolving around the continuous regime that describes some similarities with our model.

Additionally, we include a discussion for TMECor, TME, and per-player NE.

**Appendix B (Game Dynamics)**:
Due to space limitation, we moved the equations of the $\min\max$ optimization algorithms in this section and included a short discussion/description of the methods.

**Appendix C (Derivation of the Min-Max Objective in eq. 2.5)**
Since the derivation of eq. $2.5$ was not obvious, we included the calculations as to how it was derived.

**Appendix C.1**
We incorporated a short discussion and definition of the complexity class of CLS.

**Appendix E. (Proof of Theorem 3.5)**
We complemented this section with the proof for the failure of OMWU and addressed some errata of the proof for the failure of OGDA.

**Appendix F. (Proof of Theorem 3.8)**
Here we included the proof regarding the design of the matrices $K, P$.

Thank you again for your time and your insightful comments and we hope to higher support after our clarifications!

Kind regards,\
*The Authors*

---

> ### Comment · Reviewer_89w7 · 2021-11-24
> **New assumption for design matrices**
>
> I thank the authors for all their detailed answers and for already taking into account the different concerns of the reviewers in the latest version of the paper.
>
> The new Theorem 3.8 is a nice improvement about the design of the matrices $K$ and $P$. However, it seems that the condition $\beta > \alpha$ is rather strong. In particular, it does not hold if the eigenvalues of $H$ are all real. It would thus be nice to have some usual exaples where such a condition is met, to have a better understanding of this assumption.

---

> > ### Author Response · Authors · 2021-11-24
> > **New theorem 3.8**
> >
> > Dear reviewer 89w7,
> >
> > If all eigenvalues of H are real, since the trace of $H$ is zero, $H$ should have both positive and negative eigenvalues. So we agree with you that our assumption is not satisfied if all eigenvalues of $H$ are real. Nevertheless, we can show that if all eigenvalues of $H$ are real, then there are no $p,k$ with $P = p \cdot I$ and $K = k \cdot I$ so that the KPV-GDA method makes Nash equilibria stable fixed points; i.e., it is necessary to choose more complicated structure for $K,P$ when all eigenvalues of $H$ are real.
> >
> > Please also note that the Generalized Matching Pennies game we define in our manuscript induces a matrix $H$ that does not have all eigenvalues real and satisfy our assumption. It is a very interesting question to give a game theoretic interpretation of the provided assumption.
> >
> > Best,
> > The authors

---

### Decision · Program_Chairs · 2022-01-20

**Decision:**

Reject

**Comment:**

In this paper, the authors study "team-zero sum games", where two teams are facing each other with opposite objective.

The main result is that the complexity of finding equilibrium is CLS, hence probably not polynomial. This result is obtained via a reduction to some congestion games.

Three reviewers gave a mild positive score (6) while the fourth one had more concerns. I tend to agree with the first three reviewers, with a  a personal opinion around 5-6. The paper is interesting, but could benefit from polishing here and there (I acknowledge that the related work section is more precise after discussion).

This said, I also kind of agree with the last reviewer in the sense that the result of this paper is a bit narrow (also not really surprising, but we cannot always have breathtaking results), and I am also not sure that most of the ICLR community will be interested by this kind of result. This is not really a criticism, but this paper is really borderline, and this is what makes it fall into the rejection pile.

For instance, I think this paper would be more suited to some other conferences, more concerned about games and computations for instance (or even a journal).

---

> ### Public Comment · ~Ioannis_Panageas1 · 2022-01-29
> **Misunderstanding about the contribution? Learning in Games papers get accepted quite often in top ML venues**
>
> Dear Chairs,
>
> Thank you for your assessment.
>
> We would like to mention that the main result of the paper is not "the complexity of finding equilibrium is CLS, hence probably not polynomial. This result is obtained via a reduction to some congestion games.", this was quite a warm up result. The paper is about learning in games which is a research direction with many accepted submissions in top venues like ICLR, ICML, NeurIPS. Most importantly, there are quite similar settings that were indeed accepted in this ICLR (2022). As a result we are also refuting the last sentence that says "I think this paper would be more suited to some other conferences, more concerned about games and computations for instance".
>
> With all the respect, we make this comment public so that such type of judgements can be avoided in future venues. Maybe a non-expert AC,SAC should get advice from an expert AC/SAC?
>
> Best,
> IP